# P³-SAM: NATIVE 3D PART SEGMENTATION

## ABSTRACT

Segmenting 3D assets into their constituent parts is crucial for enhancing 3D understanding, facilitating model reuse, and supporting various applications such as part generation. However, current methods face limitations such as poor robustness when dealing with complex objects and cannot fully automate the process. In this paper, we propose a native 3D point-promptable part segmentation model termed P³-SAM, designed to fully automate the segmentation of any 3D objects into components. Inspired by SAM, P³-SAM consists of a feature extractor, multiple segmentation heads, and an IoU predictor, enabling interactive segmentation for users. We also propose an algorithm to automatically select and merge masks predicted by our model for part instance segmentation. Our model is trained on a newly built dataset containing nearly 3.7 million models with reasonable segmentation labels. Comparisons show that our method achieves precise segmentation results and strong robustness on any complex objects, attaining state-of-the-art performance. Our code will be released soon.

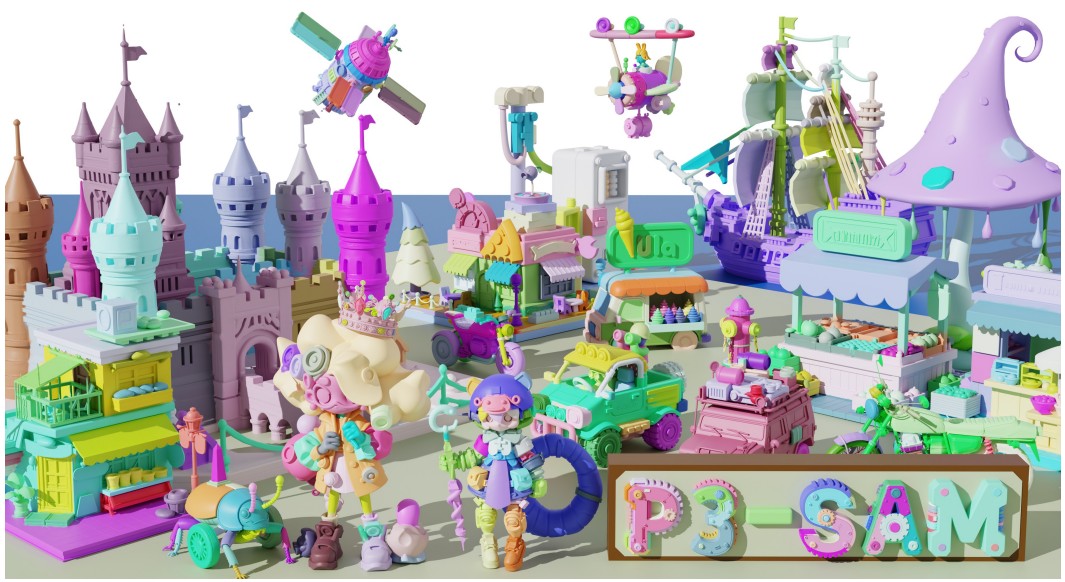

Figure 1: P³-SAM produces precise part segmentation results for any object.

## 1 INTRODUCTION

As a fundamental 3D version task, the segmentation of 3D assets plays a crucial role in shape analysis, editing, and reuse, as well as other downstream tasks such as mesh simplification and animation design Zhang et al. (2025). Many works have been proposed to address 3D segmentation and have achieved notable success, but there are still challenges in this task.

Traditional learning-based segmentation methods attempt to segment 3D point clouds using predefined part categories, relying on direct supervision from part labels. They can only segment certain parts within specific categories and struggle to handle objects with arbitrary parts. To this end, recent works Yang et al. (2024); Liu et al. (2025); Zhou et al. (2024) leverage the capabilities of the 2D SAM Kirillov et al. (2023) by lifting 2D segmentation results to serve as 3D segmentation ground truth for training 3D segmentation models. However, due to the significant data gap between 2D and 3D, these methods suffer from imprecise results and lack strong robustness when dealing with arbitrarily complex objects. Additionally, they are still one step away from full automatic segmentation, as they require users to provide the number of parts or prompt points.

In this paper, we propose a native 3D **P**oint-**P**romptable **P**art segmentation model termed $\underline{\mathbf{P}}^3$-SAM, designed to fully automate the segmentation of any complex 3D objects into components with precise mask and strong robustness. The improvements and differences of our method compared to previous methods are summarized in Table1. As a pioneering promptable image segmentation work, SAM provides a feasible implementation approach. However, our method focuses on achieving precise part segmentation automatically, and we simplify the architecture of SAM. Without adopting the complex segmentation decoder and multiple types of prompts from SAM, our model is designed to handle only one positive point prompt. Specifically, $\mathbf{P}^3$-SAM contains a feature extractor, three segmentation heads, and an IoU prediction head. We employ PointTransformerV3 Wu et al. (2024; 2025) as our feature extractor and integrate its features from different levels as extracted point-wise features. The input point prompt and feature are fused and passed to the segmentation heads to predict three multi-scale masks and an IoU predictor is utilized to evaluate the quality of the masks. To automatically segment an object, we apply our segmentation model using point prompts sampled by FPS and utilize NMSGirshick et al. (2014) to filter redundant masks. The point-level masks are then projected onto mesh faces to obtain the part segmentation results.

Another key aspect of this paper is to eliminate the influence of 2D SAM, and rely exclusively on raw 3D part supervision for training a native 3D segmentation model. While existing 3D part segmentation datasets are either too small (e.g. PartNet Mo et al. (2019)) or lack part annotation (e.g. Objaverse Deitke et al. (2022)), this work addresses the data scarcity by developing an automated part annotation pipeline for artist-created meshes and used it to generate a dataset comprising 3.7 million meshes with high-quality part-level masks. Our model demonstrates excellent scalability with this dataset and achieves robust, precise, and globally coherent part segmentation.

Our extensive experiments demonstrate that our method achieves state-of-the-art performance in part segmentation for any parts of any objects, especially on complex objects with highly detailed geometry, as shown in Figure1. The main contributions of our $P^3$-SAM are summarized as follows:

- We propose a native 3D point-promptable part segmentation model to segment any parts of any objects.

- We propose a fully automatic part segmentation approach using our model and a mask merging algorithm.

- With high accuracy, generalization, and robustness across various tasks and data types, our method can be applied to interactive, multi-head and hierarchical part segmentation.

## 2 RELATED WORK

### 2.1 TRADITIONAL 3D PART SEGMENTATION

Traditional 3D part segmentation methods usually train their networks on specific part labels from object or scene datasets, such as PartNet Mo et al. (2019), Princeton Mesh Segmentation Chen et al. (2009), ScanNet Dai et al. (2017), and S3DIS Armeni et al. (2016). These methods employ point cloud encoders like PointNet Qi et al. (2017) and PointTransformerV3 (PTv3) Wu et al. (2024) or mesh encoders like MeshCNN Hanocka et al. (2019) and Mesh Transformer (Met) Zhou et al. (2023a) to extract 3D features for the segmentation head to predict part labels. However, traditional methods suffer from limited categories and part labels and struggle to generalize to arbitrary categories and parts.

Table 1: The comparison of our method with related works across several key aspects, including the number and type of training data, the number of parameters, the time cost for full and interactive segmentation, and the ability to automatically segment objects.

| Methods | Data Num. | Data Type | Param. | Time(Seg.) | Time(Inter.) | Auto. |
|---------|-----------|-----------|--------|------------|--------------|-------|
| SAMesh | - | 2D Lifting | - | ~7min | - | ✔ |
| Find3D | 30K | 2D Data Engine | 46M | ~10s | - | ✘ |
| SAMPart3D | 200K | 2D Data Engine | 114M | ~15min | - | ✘ |
| ParField | 360K | 2D Data Engine | 106M | ~10s | - | ✘ |
| Point-SAM | 100K | 2D Data Engine | **311M** | - | ~5ms | ✘ |
| Ours | **3.7M** | **3D Native** | 112M | **~8s** | **~3ms** | ✔ |

## 2.2 2D LIFTING 3D PART SEGMENTATION

In addition to segmenting parts with semantic meaning, recent works aim to segment out geometrically significant parts. With the development of 2D foundation models, significant progress has been made by models such as CLIP Radford et al. (2021), GLIP Li et al. (2022a), SAM Kirillov et al. (2023), Dinov2 Oquab et al. (2023); Liu et al. (2024) and VLM Team (2025) in image-text alignment and zero-shot detection and segmentation. Rendering 3D models into multi-view images and leveraging these 2D foundation models for lifting 2D capabilities to 3D is an obvious but effective approach. Recent methods, such as SAMesh Tang et al. (2024), SAM3D Yang et al. (2023) and SAMPro3D Xu et al. (2023), directly apply SAM to rendered 2D images and aggregate multi-view masks to achieve class-agnostic segmentation for any 3D objects or scenes. Additionally, several methods Liu et al. (2023); Abdelreheem et al. (2023); Zhou et al. (2023b); Xue et al. (2023); Zhong et al. (2024); Umam et al. (2024); Garosi et al. (2025) utilize text descriptions of categories as prompts on 2D rendered images to enhance the querying of 3D parts. Directly lifting 2D knowledge to 3D may encounter limitations such as data gaps, 3D consistency issues, and unstable post-processing, leading to poor robustness and inaccurate segmentation results. Text-query-based methods also require prompt engineering. Rendering multi-view images and using SAM or VLM to process these images can also consume significant resources and time.

## 2.3 2D DATA ENGINE FOR 3D PART SEGMENTATION

To alleviate the 3D consistency issues and data gaps brought by directly lifting 2D knowledge, recent works Peng et al. (2023); Huang et al. (2024); Ošep et al. (2024); Ma et al. (2024) attempt to use 2D foundation models to build a data engine for training feed-forward networks on 3D point clouds and meshes. SAMPart3D Yang et al. (2024) employs a network to distill the projected Dinov2 features of point clouds. To achieve more accurate segmentation of each object, it then trains a lightweight MLP for each object to predict segmentation masks by conducting contrastive learning on SAM projections. Finally, a MLLM is utilized to annotate each part. PartField Liu et al. (2025) directly supervises a network composed of a voxel CNN and a tri-plane transformer with contrastive learning loss on both 2D and 3D masks, where the 2D masks are obtained using SAM. Point-SAM Zhou et al. (2024) adapts SAM to 3D point clouds and utilizes SAM to design a data engine based on multi-view images. This data engine continuously trains and refines a PointViT model to achieve part segmentation based on prompt points. Although a 2D data engine can reduce 3D inconsistencies and improve the network's generalization ability, segmentation based on 2D data can still suffer from boundary ambiguities and data gaps, leading to inaccurate segmentation results, especially on complex data. Additionally, these methods either require specifying the number of categories or need user-provided prompt points, which means they cannot fully automate object segmentation.

## 3 METHOD

Given the mesh $\mathbf{M} = (\mathbf{V}, \mathbf{F} \in \mathbb{N}^{N_f \times 3})$ of an object, our goal is to predict a mask $\mathbf{m}_{part} \in \{1, 2, 3, ..., N_{part}\}^{N_f}$ that segments each face into $N_{part}$ parts, where $N_f$ indicates the number of faces and $N_p$ represents the number of parts for the object. Here, each part is instance-specific but class-agnostic.

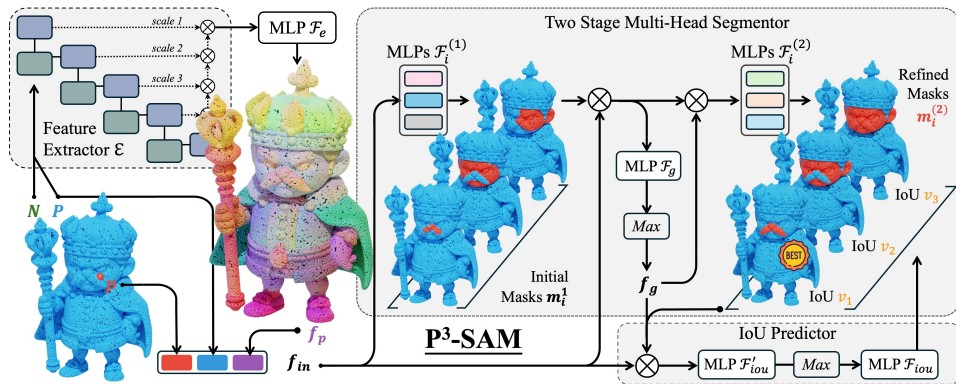

Figure 2: **The Network Architecture of P$^3$-SAM.** Input point clouds are fed to feature extractor to obtain point-wise features. The features, point prompts, and original point clouds are then fed to a two stage multi-mask segmentor to obtain three masks in various scales. Finally, the IoU predictor is utilized to evaluate the quality of the masks and select the best one as the final prediction.

## 3.1 DATA CURATION

To construct our dataset, we aggregated 3D models from multiple sources, including Objaverse Deitke et al. (2022), Objaverse-XL Deitke et al. (2023), ShapeNet Chang et al. (2015), PartNet Mo et al. (2019), and other internet repositories. We filtered models containing reasonable part information based on several simple criterions and obtained nearly 3.7 million objects. However, these object models are non-watertight. Training on such data can lead to poor generalization on watertight 3D models, such as scanned mesh or AI-generated ones. We then made nearly 2.3 million watertight models from the filtered data. During training, if a model has a watertight version, we set an 80% probability of selecting the watertight data for training. This allows our network to handle both watertight and non-watertight data.

## 3.2 POINT-PROMPTABLE PART SEGMENTATION MODEL

### 3.2.1 NETWORK ARCHITECTURE

To achieve class-agnostic part segmentation for arbitrary objects, providing prompts related to parts is more efficient than direct label supervision or contrastive learning. Previous methods utilize text as prompts to query parts, while methods like SAM Kirillov et al. (2023) and Point-SAM Zhou et al. (2024) use positive and negative prompt points or bounding boxes as prompts. To fully facilitate the automatic part segmentation of objects using point-based prompts, we design our P$^3$-SAM to segment a part using only a single point prompt. This allows the network to avoid adapting to diverse prompts, simplifying the network and improving its convergence, generalization, and accuracy. The input to our P$^3$-SAM consists of the point cloud $\mathbf{P} \in \mathbb{R}^{N_p \times 3}$ and its normals $\mathbf{N} \in \mathbb{R}^{N_p \times 3}$ sampled from the input mesh $\mathbf{M}$ and a point $\mathbf{p} \in \mathbb{R}^3$ as a prompt to indicate the part that needs to be segmented. As shown in Figure 2, the architecture of our method consists of a feature extractor, a two-stage multi-head segmentor, and an IoU predictor. Since our method only requires a single point prompt, we directly input the prompt into the segmentor.

**Feature Extractor**. Recent point cloud encoders have achieved excellent results on various point cloud tasks, especially Sonata Wu et al. (2025), a self-supervised pre-trained Point Transformer V3 Wu et al. (2024). We then employ Sonata with its pre-trained weights as our feature extractor $\mathcal{E}$ to extract multi-scale features from point clouds. We then aggregate these multi-scale features together and use an weight-shared MLP $\mathcal{F}_e$ to obtain point-wise features $f$, as shown in Figure 2, $\mathbf{f}_p = \mathcal{F}_e(\mathcal{E}(\mathbf{P}, \mathbf{N})_1, \mathcal{E}(\mathbf{P}, \mathbf{N})_2, ..., \mathcal{E}(\mathbf{P}, \mathbf{N})_n)$, where the subscripts indicate features at different scales. The point features need to be predicted only once and can be used for predicting part masks with different point prompts.

**Two-Stage Multi-Head Segmentor**. Our part dataset, introduced in Section 3.1, integrates multiple data sources which may involve varying granularity and conflicting criteria for part separation.

Additionally, the point prompts might be ambiguous in indicating the specific scale of a part, as mentioned in SAMKirillov et al. (2023). Therefore, we use a multi-head segmentor to predict multiple alternative masks at various scales in order to mitigate this conflict and ambiguity. Our multi-head segmentor contains a two-stage prediction process. In the first stage, three MLPs $\mathcal{F}_i^{(1)}$ take a mixed input $\mathbf{f}_{in}$, including the point-wise features $\mathbf{f}_p$, the input points $\mathbf{P}$ and $N_p$ copies of the point prompt $\mathbf{p}$, to predict three different masks, $\mathbf{m}_i^{(1)} = \mathcal{F}_i^{(1)}(\mathbf{f}_{in}) = \mathcal{F}_i^{(1)}(\mathbf{f}_p, \mathbf{P}, \mathbf{p}), i = 1, 2, 3$. However, the first stage is a naive implementation that lacks support for global information. Therefore, in the second stage, we introduce a global feature and re-predict the three masks based on the results from the first stage. As shown in the Figure2, we use an MLP $\mathcal{F}_g$ to predict point-wise features and apply max pooling along the point dimension. We utilize an MLP $\mathcal{F}_g$ to predict point-wise features, and apply max pooling along the point dimension to derive a global feature, $\mathbf{f}_g = MaxPool(\mathcal{F}_g(\mathbf{f}_{in}, \mathbf{m}_1^{(1)}, \mathbf{m}_2^{(1)}, \mathbf{m}_3^{(1)}))$. Finally, three new MLPs $\mathcal{F}_i^{(2)}$ are employed to predict more accurate results based on the global features and the outcomes from the first phase, $\mathbf{m}_i^{(2)} = \mathcal{F}_i^{(2)}(\mathbf{f}_{in}, \mathbf{f}_g, \mathbf{m}_1^{(1)}, \mathbf{m}_2^{(1)}, \mathbf{m}_3^{(1)}), i = 1, 2, 3$. While the second stage optimizing the results from the first stage, the initial masks $\mathbf{m}_i^{(1)}$ also help the second stage to focus the extraction of global features on the parts that need segmentation, making feature extraction more efficient and improves the accuracy of segmentation results.

**IoU Predictor**. To achieve automatic identification of the best mask, we introduce an IOU predictor to assess the quality of $\mathbf{m}_1^{(2)}, \mathbf{m}_2^{(2)}, \mathbf{m}_3^{(2)}$ and select the best mask as the network's final prediction. The assessment is achieved by directly predicting the IoU values of the predicted masks and the ground truth masks. The IOU predictor first uses an MLP $\mathcal{F}_{iou}'$ and max pooling to obtain a global feature from the global feature and three masks of second stage, and then employs another MLP $\mathcal{F}_{iou}$ to predict three IoU values, $\mathbf{v}_1, \mathbf{v}_2, \mathbf{v}_3 = \mathcal{F}_{iou}(MaxPool(\mathcal{F}_{iou}'(\mathbf{f}_{in}, \mathbf{f}_g, \mathbf{m}_1^{(2)}, \mathbf{m}_2^{(2)}, \mathbf{m}_3^{(2)})))$. The two-stage multi-head segmentor and IoU predictor are lightweight models capable of real-time computation. Consequently, once the global feature of a given 3D model is extracted, our P$^3$-SAM can be utilized for real-time interactive segmentation.

### 3.2.2 TRAINING

**Data augmentation.** To enhance the robustness of our network, we introduce random noise to the input points $\mathbf{P}$, normals $\mathbf{N}$, and point prompts $\mathbf{p}$ during training. Furthermore, we randomly remove normals with a probability of 0.3. We also mix watertight and non-watertight data, as mentioned in Section 3.1.

**Optimization Losses.** During training, for a given model, we randomly select $K$ part masks and then randomly choose one point from the part points corresponding to each mask, resulting in $K$ prompts $\mathbf{p}_j \in \mathbb{R}^3$ and $K$ ground truth part masks $\mathbf{m}_j^{(gt)} \in \{0, 1\}^{N_P}$, where $j = 1, 2, ..., K$. For the three masks generated by the network in the first and second stages, we apply both Dice loss $\mathcal{L}_{dice}$ and Focal loss $\mathcal{L}_{focal}$ for supervision. Backpropagation is applied only to the output with the lowest loss, which encourages each segmentation head to predict masks at different scales. So, the mask loss $\mathcal{L}_{mask}$ can be calculated as:

$$\mathcal{L}_{mask}^{(t)} = \frac{1}{K} \sum_{j=1}^{K} \min_{i=1}^{3} \left( \alpha_{dice} \mathcal{L}_{dice}(\mathbf{m}_{ij}^{(t)}, \mathbf{m}_j^{(gt)}) + \mathcal{L}_{focal}(\mathbf{m}_{ij}^{(t)}, \mathbf{m}_j^{(gt)}) \right),$$

where $\alpha_{dice}$ is a weighting parameter, and $t = 1, 2$ indicates whether the loss is computed for the first or second stage of the network. To supervise the IoU, we first calculate the IoU between $\mathbf{m}_{ij}^{(2)}$ and $\mathbf{m}_j^{(gt)}$. We then use MSE to compute the loss based on these IoU values,

$$\mathcal{L}_{IoU} = \frac{1}{3K} \sum_{j=1}^{K} \sum_{i=1}^{3} \mathcal{L}_{MSE} \left( \mathbf{v}_{ij}, IoU(\mathcal{I}(\mathbf{m}_{ij}^{(2)}), \mathbf{m}_j^{(gt)}) \right),$$

where $\mathcal{I}$ is an indicator function that indicates whether the mask value is greater than 0.5. The overall loss is the sum of the mask losses from both the first and second stages and the IOU loss, that is $\mathcal{L} = \mathcal{L}_{mask}^{(1)} + \mathcal{L}_{mask}^{(2)} + \mathcal{L}_{IoU}$.

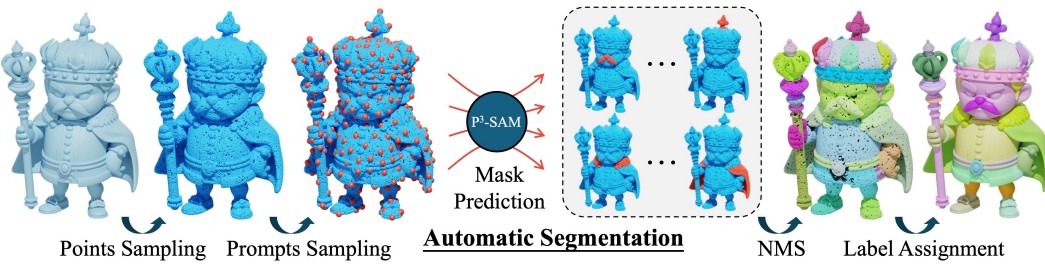

Points Sampling    Prompts Sampling    **Automatic Segmentation**    NMS    Label Assignment

Figure 3: **Automatic Segmentation Pipeline.** Point prompts are sampled by FPS and go through the P³-SAM to obtain multiple masks. NMS is then adopted to merge redundant masks. The point-level masks are then projected onto mesh faces to obtain the part segmentation results.

Table 2: The comparison of our method with previous methods on PartObj-Tiny. The first two blocks represent class-agnostic part segmentation without and with connectivity, respectively, and the last block represents interactive segmentation.

| Task | Method | Human | Animals | Daily | Build. | Trans. | Plants | Food | Elec. | AVG. |
|---|---|---|---|---|---|---|---|---|---|---|
| | Find3D | 23.99 | 23.99 | 22.67 | 16.03 | 14.11 | 21.77 | 25.71 | 19.83 | 21.28 |
| Seg. w/o | SAMPart3D | 55.03 | 57.98 | 49.17 | 40.36 | 47.38 | 62.14 | **64.59** | 51.15 | 53.47 |
| Connect. | PartField | 54.52 | 58.07 | 56.46 | 42.47 | 49.09 | 59.16 | 55.4 | 56.29 | 53.93 |
| | Ours | **60.77** | **59.43** | **62.98** | **50.82** | **57.72** | **70.53** | 54.04 | **61.96** | **59.88** |
| | SAMesh | 66.03 | 60.89 | 56.53 | 41.03 | 46.89 | 65.12 | 60.56 | 57.81 | 56.86 |
| Seg. w/ | PartField | 80.85 | 83.43 | 77.83 | **69.66** | **73.85** | 80.21 | 85.27 | **82.30** | 79.18 |
| Connect. | Ours | 80.77 | **86.46** | **80.97** | 67.77 | 68.44 | **90.30** | **92.90** | 81.52 | **81.14** |
| Interact. | Point-SAM | 26.13 | 29.25 | 28.85 | 23.58 | 22.91 | 31.44 | 33.04 | 28.05 | 27.91 |
| | Ours | **49.01** | **53.45** | **52.36** | **38.50** | **51.52** | **62.57** | **50.80** | **51.86** | **51.23** |

## 3.3 AUTOMATIC SEGMENTATION

Methods such as interactive segmentation, text prompt extraction, and clustering often require human intervention during the segmentation process. To achieve fully automatic segmentation, we propose an automated approach based on our P³-SAM, as shown in Figure 3. Our method begins by sampling points $\mathbf{P}$ and normals $\mathbf{N}$ from given mesh $\mathcal{M}$. Subsequently, we use Farthest Point Sampling (FPS) to select $N_{pp}$ point prompts $\mathbf{p}_j$ from $\mathbf{P}$. After extracting features $\mathbf{f}_p$ from $\mathbf{P}$, we predict a mask $\mathbf{m}_j$ and an IoU value $\mathbf{v}_j$ based on each point prompt $\mathbf{p}_j$ utilizing our P³-SAM. To ensure that each part can be segmented out, point prompts are always over-sampled, with their number being significantly greater than the actual number of parts in the object. To obtain the true number of parts, we use Non-Maximum Suppression (NMS) to filter out the numerous duplicate masks. We first sort the masks in descending order according to their IoU values to form a candidate queue. We take out the first mask and use it to filter out other masks in the queue that have an IoU greater than $T_{NMS}$ with this mask. We repeat this process of selecting and filtering until no masks remain in the queue. The set of all selected masks constitutes the final result of NMS. The part number $N_{part}$ is the number of the selected masks, and each mask has its own part label. According to which face and mask each point belongs to, we assign the corresponding part labels to each face and determine a final part label for each patch through voting. We use the flood fill algorithm to assign labels to faces that do not have a label. Specifically, for each unlabeled face, we assign it the most frequent label among its neighboring faces (or its nearest several faces if there is no connectivity in the mesh). We repeat this process until all faces have been assigned a label and obtain the final mask $\mathbf{m}_{part}$ of each faces in mesh $\mathcal{M}$.

## 4 EXPERIMENTS

### 4.1 COMPARISON

**Evaluation Datasets.** We evaluate each method on three datasets: PartObj-Tiny Yang et al. (2024), PartObj-Tiny-WT, and PartNetE Liu et al. (2023). PartObj-Tiny-WT is the watertight version of PartObj-Tiny for evaluation of the various networks on watertight data.

Table 3: The comparison of various methods on PartObj-Tiny-WT.

| Task | Fully Segmentation w/o Connectivity | | | | | Interactive Seg. | |
|---|---|---|---|---|---|---|---|
| Method | Find3D | SAMPart3D | SAMesh | PartField | Ours | Point-SAM | Ours |
| PartObj-Tiny-WT | 20.76 | 48.79 | - | 51.54 | **58.10** | 24.16 | **49.11** |
| PartNetE | 21.69 | 56.17 | 26.66 | 59.1 | **65.39** | 45.85 | **63.48** |

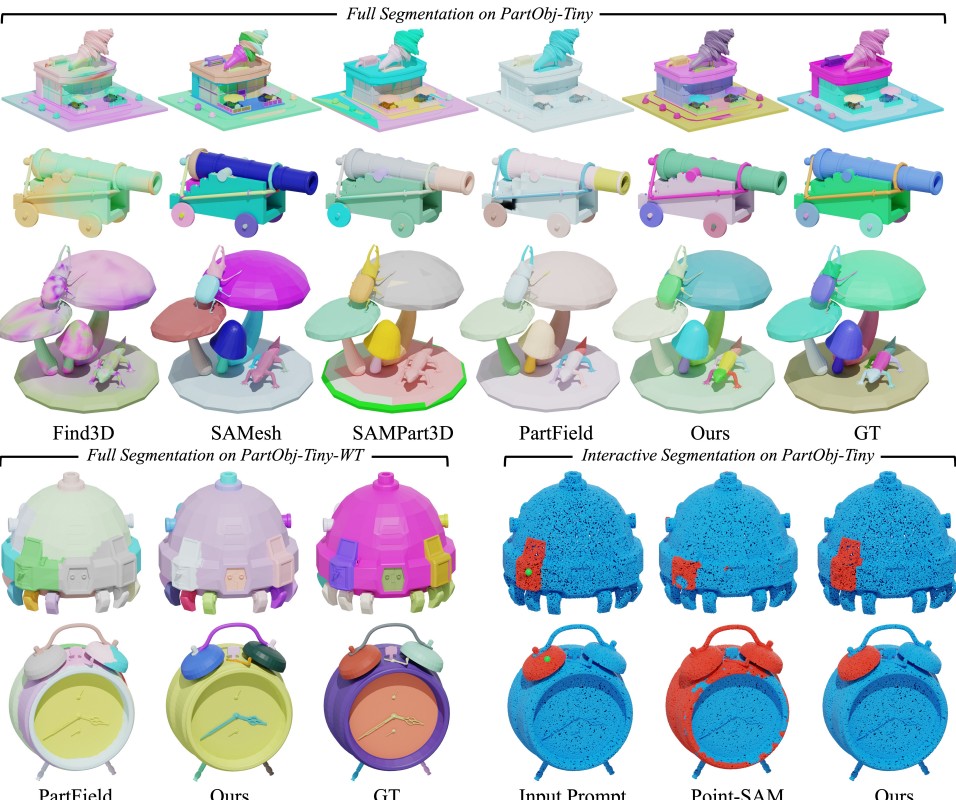

*Full Segmentation on PartObj-Tiny*

Find3D    SAMesh    SAMPart3D    PartField    Ours    GT

*Full Segmentation on PartObj-Tiny-WT*    *Interactive Segmentation on PartObj-Tiny*

PartField    Ours    GT    Input Prompt    Point-SAM    Ours

Figure 4: The comparison of our method across different tasks.

**Tasks.** There are three tasks: full segmentation without connectivity, full segmentation with connectivity, and interactive segmentation. The meshes in PartObj-Tiny are non-watertight, with each face exhibiting strong connectivity, forming distinct connected components that are strongly correlated with the segmentation results. Introducing such connectivity may improve segmentation performance. However, in real-world applications, the connectivity relationships of meshes are often chaotic or may not even exist. We then divide the full segmentation task into the first two tasks. For watertight data and point clouds, there is no connectivity, so the second task does not apply.

**Baseline Methods.** We compare our P³-SAM with recent related works including SAMesh Tang et al. (2024), Find3D Ma et al. (2024), SAMPart3D Yang et al. (2024), ParField Liu et al. (2025) and Point-SAM Zhou et al. (2024). More comparisons on different aspects, including time cost, number of parameters, amount of training data, etc., are shown in Table 1.

**Metric.** The evaluation metric for fully segmentation is the same as in previous work Liu et al. (2025), using IoU to measure the accuracy of mask predictions. To evaluate the interactive segmentation, we sample 10 prompt points for each part, then measure the average IOU between the predicted masks for all prompts of all parts and their corresponding ground truth masks.

**Results.** Table 2 shows the evaluation results of various methods on PartObj-Tiny across three tasks. In the second task, the results for PartField Liu et al. (2025) are based on its original version, which incorporates connected components as the basis for hierarchical clustering and selects the optimal results from multiple levels. To ensure a fair comparison, we also introduced connected components

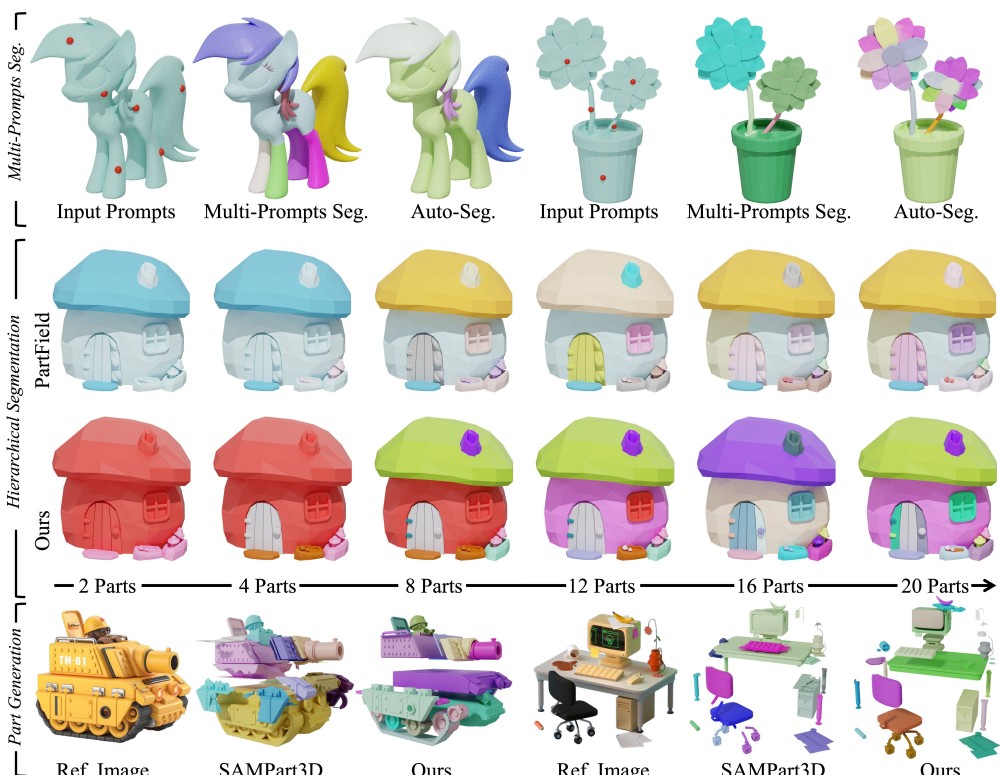

Figure 5: The three applications of our method.

and used random prompts for each part. The detailed methodology can be found in Section A.5. In the third task, since Point-SAM can only segment point clouds, we sample the point clouds from the meshes for comparison. Note that our method is also capable of segmenting point clouds because we do not require the connectivity of the mesh. The comparisons on PartObj-Tiny-WT and PartNetE are shown in Table 3. Since watertight data and point clouds lack connectivity information, PartField's performance is not as good as on non-watertight data. This again validates that our method effectively learns the geometric features of objects. We also conduct a qualitative comparison of our method and previous methods, as shown in Figure 4. Quantitative and qualitative comparison across various datasets and tasks, involving different data forms such as non-watertight meshes, watertight meshes, and point clouds, demonstrate the remarkable effectiveness, robustness and generalization ability of our methods, confirming its superior performance under diverse conditions.

## 4.2 APPLICATIONS

**Multi-Prompts Auto-Segmentation.** Our method can also segment the object given several point prompts that indicate specific parts. As shown in Figure 5, the multi-prompts segmentation can follow the user's instructions. Compared to automatic segmentation, it can both extend unsegmented regions, such as the horse's body, and merge over-segmented regions, such as flower petals.

**Hierarchical Part Segmentation.** As shown in Figure 5, our hierarchical segmentation results effectively aggregate different parts at various levels, validating the effectiveness of our feature extraction method in accurately representing the information of each part. Compared to the results from PartField, our method's aggregation better adheres to the relationships between parts.

**Part Generation.** Our results can also be applied to part generation Yang et al. (2025b); Yan et al. (2025) as an instruction for splitting objects. Figure 5 demonstrates the exploded part generation results of HoloPart Yang et al. (2025a) when given the segmentation masks from SAMPart3D Yang et al. (2024) and ours. Our more accurate segmentation masks can significantly improve HoloPart's performance, helping it generate cleaner and more precise parts.

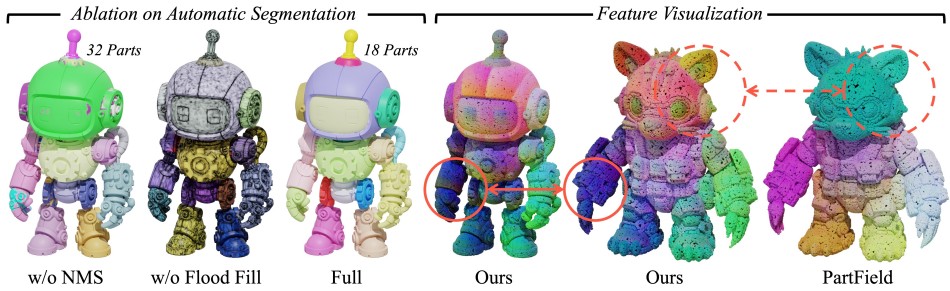

Figure 6: The ablation study of our method and the visualization of features.

Table 4: The comparison of four ablated methods and our full method on the test set of our dataset.

| Ablation | w/o Augmentation | | | | w/ Augmentation |
| --- | --- | --- | --- | --- | --- |
| | Single-Head | Stage 1 Only | Stage 2 Only | Stage 1 + Stage 2 | Full |
| mIoU | 0.2801 | 0.4265 | 0.6647 | 0.7464 | **0.7906** |

## 4.3 ABLATION STUDY

We first conduct ablation studies on our network architecture. The feature extractor can be any point cloud encoder, and we select the current state-of-the-art one that is Sonata. The IoU predictor is critical for selecting the best mask; without it, our method cannot function properly. We then focus on conducting ablation studies of our two-stage multi-head segmentor. As shown in Table 4, we evaluate four ablated models and our full method on the test set of our dataset. The first model contains only one segmentation head of the first stage. The second and third models respectively include only the first and second stages. The fourth model includes both stages but is trained without data augmentation. The last model is our full version. This progressive ablation study clearly demonstrates the importance of each component in our method. Notably, the difference between the second and third models is that the latter extracts a global feature during segmentation. The better performance metrics of the third model highlight the importance of this global feature. As shown in Figure 6, we also present the full segmentation results without using the NMS or flood fill algorithm in our automatic segmentation approach. The results highlight the necessity of these two steps, as without them, the segmentation masks are not complete, have unclear boundaries, and do not yield a reasonable number of parts. Figure 6 also visualizes the point-wise features of objects. The results show that for the same type of data, our method produces similar features for corresponding parts, while our features can capture more detailed geometry compared to PartField, such as the eyes and ears of the person on the right. This fully demonstrates the advantages of our method in extracting accurate features.

## 5 LIMITATIONS AND CONCLUSIONS

In this paper, we propose P$^3$-SAM, a native 3D part segmentation method. Our approach employs Sonata to extract point-wise features and uses a two-stage multi-head segmentor to predict multi-scale masks given a point prompt indicating a part. An IoU predictor is employed to evaluate and select the best mask. We also propose an automatic segmentation approach using our P$^3$-SAM. We train our model on 3.7 million models, resulting in a part segmentation method with high accuracy, generalization, and robustness across various tasks and data types. Our method is also flexible and can be applied to multiple applications such as real-time interactive, hierarchical, or multi-prompt part segmentation. We observe that our method may rely too heavily on the geometric information of the object's surface and lacks an understanding of the spatial volume of the object's parts. This is because our training data consists solely of surface point clouds. Therefore, future work may focus on developing models with spatial segmentation capabilities to broaden their applicability to a wider range of tasks.

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

# A  APPENDIX

## A.1  THE USE OF LARGE LANGUAGE MODELS (LLMS)

All technical contributions, including the methodology, equations, and results, are solely the work of the authors.

## A.2  MORE RELATED WORKS DISCUSSION

In this section, we provide a more detailed introduction to related works and highlight the differences between these existing methods and our proposed method.

### A.2.1  TRADITIONAL 3D PART SEGMENTATION

Traditional 3D part segmentation methods usually train their networks on specific part labels from object or scene datasets, such as PartNet Mo et al. (2019), Princeton Mesh Segmentation Chen et al. (2009), ScanNet Dai et al. (2017), and S3DIS Armeni et al. (2016). These methods employ point cloud encoders like PointNet Qi et al. (2017) and PointTransformerV3 (PTv3) Wu et al. (2024) or mesh encoders like MeshCNN Hanocka et al. (2019) and Mesh Transformer (Met) Zhou et al. (2023a) to extract 3D features for the segmentation head to predict part labels.

These methods require part labels with clear categories for training. However, labeling such detailed information on a large scale is impractical, leading to traditional methods suffering from limited categories and part labels, and struggling to generalize to arbitrary categories and parts. In this paper, we focus more on class-agnostic part instance segmentation for arbitrary objects. Therefore, traditional methods are not within our scope of consideration and comparison.

### A.2.2 2D LIFTING 3D PART SEGMENTATION

With the development of 2D foundation models, significant progress has been made by models such as CLIP Radford et al. (2021), GLIP Li et al. (2022a), SAM Kirillov et al. (2023), Dinov2 Oquab et al. (2023) and VLM in image-text alignment and zero-shot detection and segmentation. Rendering 3D models into multi-view images and leveraging these 2D foundation models for lifting 2D capabilities to 3D is an obvious but effective approach. Recent methods, such as SAMesh Tang et al. (2024), SAM3D Yang et al. (2023) and SAMPro3D Xu et al. (2023), directly apply SAM to rendered 2D images and aggregate multi-view masks to achieve class-agnostic segmentation for any 3D objects or scenes. Additionally, several methods utilize text descriptions of categories as prompts on 2D rendered images to enhance the querying of 3D parts. PartSLIP Liu et al. (2023) and SATR Abdelreheem et al. (2023) employ text prompts and GLIP Li et al. (2022a) to detect parts, followed by post-processing to segment parts on point clouds and meshes. Besides GLIP, PointSLIP++ Zhou et al. (2023b) and ZeroPS Xue et al. (2023) also leverage SAM, achieving more precise segmentation results. The MeshSegmenter Zhong et al. (2024) employs Stable Diffusion Rombach et al. (2022) to generate textures for a mesh, enabling SAM Kirillov et al. (2023) and Grounding DINO Liu et al. (2024) to clearly segment and detect parts. PartDistill Umam et al. (2024) utilizes 2D Vision-Language Models for forward and backward knowledge distillation to achieve 3D part segmentation. COPS Garosi et al. (2025) leverages Dinov2 Oquab et al. (2023) to project visual features onto 3D point clouds and then uses a VLM for part segmentation.

Directly lifting 2D knowledge to 3D may encounter limitations such as data gaps, 3D consistency issues, and unstable post-processing, leading to poor robustness and inaccurate segmentation results. Text-query-based methods also require prompt engineering. Different from these methods, our native 3D method directly processes and trains on 3D objects, avoiding the introduction of 2D data and eliminating the aforementioned data gap. Besides, all these methods require rendering 3D models. For instance, SAMesh requires rendering 12 images at the vertices of an icosahedron. Rendering multi-view images and using SAM or VLM to process these images can also consume significant resources and time. Again, our native 3D method does not require rendering images, and its inference speed is significantly faster than these methods.

### A.2.3 2D DATA ENGINE FOR 3D PART SEGMENTATION

To alleviate the 3D consistency issues and data gaps brought by directly lifting 2D knowledge, recent works attempt to use 2D foundation models to build a data engine for training feed-forward networks on 3D point clouds and meshes. OpenScene Peng et al. (2023) employs a feature extractor to learn the CLIP features projected onto scene point clouds and uses text prompts to query these features for segmentation. Segment3D Huang et al. (2024) and SAL Ošep et al. (2024) use feed-forward networks to learn the projected masks of scene meshes and LiDAR data predicted by SAM, and then use CLIP to assign categories to each part. Find3D Ma et al. (2024) trains a network to segment objects given text prompts by building a data engine that allows the VLM to query parts after SAM processes multi-view images. SAMPart3D Yang et al. (2024) employs a network to distill the projected Dinov2 features of point clouds. To achieve more accurate segmentation of each object, it then trains a lightweight MLP for each object to predict segmentation masks by conducting contrastive learning on SAM projections. Finally, a MLLM is utilized to annotate each part. PartField Liu et al. (2025) directly supervises a network composed of a voxel CNN and a tri-plane transformer with contrastive learning loss on both 2D and 3D masks, where the 2D masks are obtained using SAM. Point-SAM Zhou et al. (2024) adapts SAM to 3D point clouds and utilizes SAM to design a data engine based on multi-view images. This data engine continuously trains and refines a PointViT model to achieve part segmentation based on prompt points.

Although a 2D data engine can reduce 3D inconsistencies and improve the network's generalization ability, segmentation based on 2D data can still suffer from boundary ambiguities and data gaps, leading to inaccurate segmentation results, especially on complex data. Additionally, these meth-

ods either require specifying the number of categories or need user-provided prompt points, which means they cannot fully automate object segmentation. Find3D and SAMPart3D utilize rendered 2D images of objects to build their data engines, while PartField and Point-SAM also leverage several 3D part segmentation datasets such as PartNet Mo et al. (2019) and ScanNet Dai et al. (2017). However, the availability of native 3D data is limited, which restricts the improvement in segmentation performance. On the contrary, our method is trained exclusively on native 3D data, and the quantity of data we use is significantly larger than that of other methods. As a result, our approach can more fully learn the geometric features of 3D objects and achieve better segmentation performance.

The difference between our method and PartField is that PartField uses a clustering approach after extracting the features of the input, whereas our method employs a segmentation module and point prompts.

The difference between our method and Point-SAM lies in the following aspects:

- **Motivation:** While PointSAM focuses primarily on interactive segmentation tasks, our method is specifically designed for automatic full segmentation. PointSAM does not provide a comprehensive solution for full segmentation, whereas our approach aims to deliver an efficient and effective method for automatically segmenting entire datasets without requiring user interaction.

- **Network Architecture:** PointSAM adapts the network structure of SAM to build a model suitable for point clouds. It utilizes FPS to extract key points and groups surrounding points as tokens, directly employing SAM's architecture as the segmentation network. However, extending 2D network structures to 3D is not always effective. Therefore, unlike PointSAM, we employ state-of-the-art methods for point cloud feature extraction as our feature extraction module. To simplify model training and inference, we removed the complex prompt encoding and adopted a single prompt point. To enhance mask quality, we also introduced a two-stage segmentation module.

- **Data:** Our method does not have a data engine; we directly train on our large-scale preprocessed native 3D data. In contrast, PointSAM constructs a data engine that leverages prior knowledge from 2D SAM, which inevitably introduces inconsistencies from 2D.

- **Our Advantages:**
  - Our method is a fully native 3D segmentation approach that supports both interactive segmentation and automatic full segmentation.
  - Our method is trained on a large-scale preprocessed native 3D dataset of 3.7 million samples, avoiding the introduction of 2D segmentation priors.
  - Our method exhibits strong generalization, robustness, and efficiency, outperforming existing methods and achieving state-of-the-art results across multiple datasets and tasks.

## A.3 MORE METHOD DETAILS

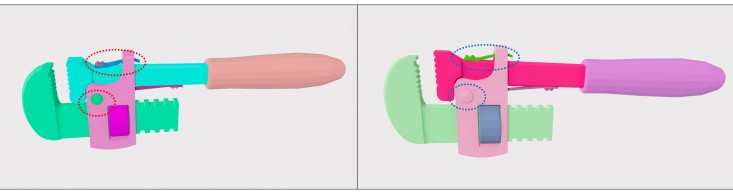

Figure 7: Example of part merge.

### A.3.1 DATA CURATION DETAILS

To construct our dataset, we aggregated 3D models from multiple sources, including Objaverse Deitke et al. (2022), Objaverse-XL Deitke et al. (2023), ShapeNet Chang et al. (2015), PartNet Mo et al. (2019), and other internet repositories. The construction of our dataset adheres to a few

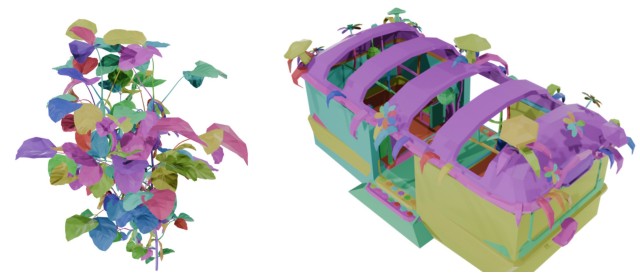

Figure 8: Examples with too many parts (>50).

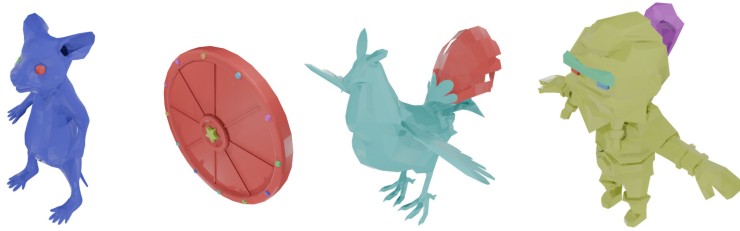

Figure 9: Examples with imbalanced parts, the largest part cover more than 85% of the surface area.

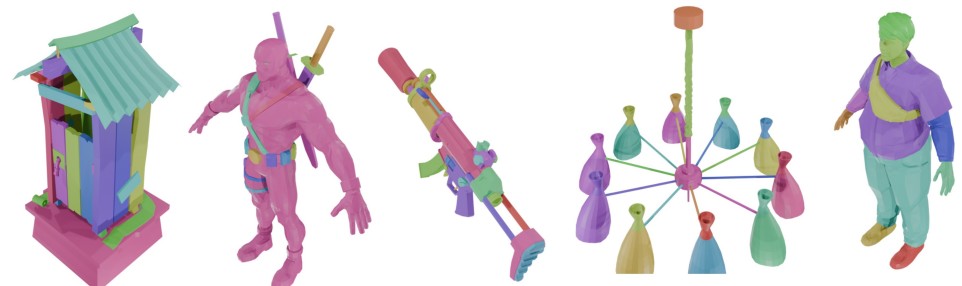

Figure 10: Example of valid data.

simple principles and filtering criteria mentioned in the paper, and it does not involve any manual intervention, significantly reducing human labor. The specific data processing and filtering pipeline is as follows:

1. **Checking for Connected Components.** The 3D assets are primarily created by artists, who craft 3D shapes part by part and then assemble them. Since the assembly process often does not merge meshes of parts, we can reverse-engineer the part information from the asset. This component-based construction method can be leveraged to obtain component-wise segmentation data, which forms an indispensable foundation for building our dataset.

2. **Compute Area and Occupied Voxels.** Specifically, the complete mesh is first decomposed into sub-meshes based on connected components. We calculate the area of each connected component's triangular facets and their overall proportion. Then, we divide the space into voxels with a resolution of 128 and compute the occupied voxels for each component. When two components have overlapping occupied voxels, they are considered adjacent, and an adjacency matrix is constructed. This step forms the basis for subsequently merging extremely small connected components.

3. **Merge Extremely Small Connected Components.** For each connected component, if its area proportion is less than 1%, it is merged with the largest adjacent part. If there are no adjacent parts, no action is taken. This step is repeated until all possible merges are completed. This process helps avoid mask distribution imbalances caused by extremely small parts, thereby improving the utilization of 3D models. A large number of small

parts can generate many masks without significant actual area, which can negatively impact training. Without merging, much of the data would be unusable. Additionally, very small parts are not conducive to subsequent watertight processing. Since the merged parts are relatively small, they have minimal impact on the overall quality of most masks. Although some low-quality erroneous masks may still be generated, we believe that the large volume of data can mitigate the effects of these noisy data. (Figure 7)

4. **Filter Out Data with Too Few or Too Many Components.** After merging, we filter out data where the number of parts is less than 2 or greater than 50. Data containing only a single part do not involve object segmentation and need to be filtered out. Data with an excessive number of parts often lack reasonable semantic meaning for each part and can lead to over-segmentation, resulting in a large number of unnecessary parts. For example, a car data containing thousands of parts typically consist of surface patches that do not have meaningful semantic significance. (Figure 8)

5. **Filter Out Imbalanced Data.** When an object contains a very large part, i.e., one part occupies more than 85% of the total area, we filter out this object. Additionally, if an object contains a large number of independent small parts, where the total area of parts with less than 1% area proportion (which could not be merged in step 3) exceeds 10%, we also filter out this object. This step is aimed at removing imbalanced masks, specifically those with extremely large actual regions and those with extremely small actual regions. (Figure 9)

The primary purpose of our data processing and filtering is to improve data utilization and generate balanced mask distributions. After the aforementioned filtering steps, we obtain nearly 3.7 million objects, as shown in Figure 10. During training, we sample points $\mathcal{P}_{nwt}$ from the meshes and record which part each point was sampled from. Therefore, the ground truth label $\mathcal{L}_{nwt}$ for each sampled point is its corresponding part label.

However, these object models are non-watertight at the object-level, often containing internal structures and clear boundaries. Training solely on such data can lead to poor generalization on watertight 3D models, such as scanned mesh or AI-generated ones. We then made the filtered models watertight Hunyuan3D (2025), resulting in nearly 2.3 million successfully watertight models. These watertight models do not contain internal structures and only include the outer surfaces of the models. To obtain the ground truth labels $\mathcal{L}_{wt}$ of the points $\mathcal{P}_{wt}$ sampled from the watertight meshes of an object in our dataset, we follow these steps. First, we sample points $\mathcal{P}_{nwt}$ from the non-watertight mesh of the object, along with their corresponding ground truth labels $\mathcal{L}_{nwt}$. Simultaneously, we sample points $\mathcal{P}_{wt}$ from the watertight mesh of the same object. For each point $\mathbf{p}_{wt}$ in $\mathcal{P}_{wt}$, we find its nearest neighbor $\mathbf{p}_{nwt}$ in $\mathcal{P}_{nwt}$. We then assign the label of $\mathbf{p}_{nwt}$ to $\mathbf{p}_{wt}$, ensuring that each point in $\mathcal{P}_{wt}$ has an accurate ground truth label $\mathcal{L}_{wt}$. During training, if a model has a watertight version, we set an 80% probability of selecting the watertight data for training. This allows our network to handle both watertight and non-watertight data.

### A.3.2 DATA AUGMENTATION DETAILS

We first set the maximum scale of the noise $s_{max}$ to 0.01. For the augmentation of input points $\mathbf{P}$ and normals $\mathbf{N}$, we randomly select a scale $s$ from $(0, s_{max})$ to simulate varying levels of point cloud noise, thereby making our method more robust. Then, the points $\mathbf{P}$ and normals $\mathbf{N}$ are augmented with noise as follows:

$$\mathbf{P}' = \mathbf{P} + \mathcal{N}(0, s),$$

$$\mathbf{N}'' = \frac{\mathbf{N}'}{||\mathbf{N}'||}, \mathbf{N}' = \mathbf{N} + \mathcal{N}(0, s) * 10.$$

The augmentation of prompt $\mathbf{p}$ is

$$\mathbf{p}' = \mathbf{p} + \mathcal{N}(0, s_{max}).$$

### A.3.3 AUTOMATIC SEGMENTATION DETAILS

Our automatic segmentation approach is shown in Algorithm 1 and the NMS for masks is shown in Algorithm 2.

---

**Algorithm 1** Automatic Segmentation

---

**Input:** Mesh $\mathcal{M}$ with $N_f$ faces
**Output:** Mask $\mathbf{m}_{part} \in \{1, 2, ..., N_{part}\}^{N_f}$ with $N_{part}$ parts
1: Sample $N_p$ points $\mathbf{P}$ with normals $\mathbf{N}$ from $\mathcal{M}$
2: Sample $N_{pp}$ prompt points $\mathbf{p}_j$ from $\mathbf{P}$ using FPS
3: Extract point-wise feature $\mathbf{f}_p$ from $\mathbf{P}$ and $\mathbf{N}$ using P$^3$-SAM
4: Predict $N_{pp}$ masks $\mathbf{m}_j$ and IoU values $\mathbf{v}_j$ based on $\mathbf{p}_j$ using P$^3$-SAM
5: Filter masks $\mathbf{m}_j$ using NMS and retain $N_{part}$ masks
6:     Sort the masks $\mathbf{m}_j$ and their corresponding IoU values $\mathbf{v}_j$ in descending order based on the IoU values.
7: Assign the $N_{part}$ point masks $\mathbf{m}_j$ to $\mathcal{M}$ with part labels
8: Fill the faces without labels using flood fill algorithm and produce the mask $\mathbf{m}_{part}$

---

---

**Algorithm 2** NMS

---

**Input:** Masks $\mathbf{m} = \{\mathbf{m_1}, \mathbf{m_2}, ..., \mathbf{m_n}\}$ and their corresponding IoU values $\mathbf{v} = \{\mathbf{v_1}, \mathbf{v_2}, ..., \mathbf{v_n}\}$
**Output:** Masks $\mathbf{m} = \{\mathbf{m_1}, \mathbf{m_2}, ..., \mathbf{m_k}\}$
1: Sort the masks $\mathbf{m}$ in descending order based on the IoU values $\mathbf{v}$.
2: **for** Mask $\mathbf{m_i}$ in masks $\mathbf{m}$ **do**
3:   **for** Other mask $\mathbf{m_j}$ in masks $\mathbf{m}$ **do**
4:     **if** $IoU(\mathbf{m_i}, \mathbf{m_j}) > 0.9$ **then**
5:       Remove mask $\mathbf{m_j}$
6:     **end if**
7:   **end for**
8: **end for**

---

### A.3.4 IMPLEMENTATION DETAILS

To better handle complex objects, we reduced the voxel size of the input to Sonata. The channel of the point-wise feature $\mathbf{f}_p$ is 512. The point number $N_p$ is set to $100,000$ during training, evaluation, and inference. We randomly select $K = 8$ parts in the training process and set $\alpha_{dice}$ to 0.5. For automatic segmentation, we sample $N_{pp} = 400$ prompts from points, and the threshold $T_{NMS}$ is set to 0.9. Our network is trained on our dataset using 64 H20 for 9 epochs. We set the batch size to 2 per GPU, and the training took approximately 4 days. We employ the Adam optimizer with a learning rate of $10^{-5}$.

### A.4 EVALUATION DETAILS

**Evaluation Datasets Details.** We evaluate each method on three datasets: PartObj-Tiny Yang et al. (2024), PartObj-Tiny-WT, and PartNetE Liu et al. (2023). PartObj-Tiny is a subset of Objarvse Deitke et al. (2022), containing 200 data samples across 8 categories, with manually annotated part segmentation information. PartObj-Tiny-WT is the watertight version of PartObj-Tiny. To evaluate the performance of various networks on watertight data, we converted the meshes from PartObj-Tiny to watertight versions and successfully obtained 189 watertight meshes. We then acquired the ground truth segmentation labels following the method described in Section A.3.1. And there is no color on the watertight mesh, which could pose a challenge for methods based on rendering multi-view images. PartNetE, derived from PartNet-Mobility, contains 1,906 shapes covering 45 object categories in the form of point clouds. We also evaluate various networks on it to verify their generalization performance on point cloud.

**Baseline Methods Details.** We compare our P$^3$-SAM with recent related works including SAMesh Tang et al. (2024), Find3D Ma et al. (2024), SAMPart3D Yang et al. (2024), ParField Liu et al. (2025) and Point-SAM Zhou et al. (2024). Among these, SAMesh is a method based on 2D lifting that enables fully automatic segmentation. The other methods based on 2D data engines require human intervention in the overall segmentation process: Find3D requires text prompts, SAMPart3D and ParField need part categories for clustering, and Point-SAM necessitates manual selection of prompt points. Since Point-SAM cannot segment the entire model, we only compare its performance on interactive segmentation.

**More Results Analysis.** Table 2 shows the evaluation results of various methods on PartObj-Tiny across three tasks. Although PartField Liu et al. (2025) performs well in the second task, once connectivity information is removed, its performance drops significantly, indicating that the PartField method is not robust to meshes without connected components. This is further evidenced by the comparison on watertight data, where the absence of connectivity also affects its performance. In contrast, our method consistently achieves the best performance regardless of whether connectivity information is present or not. This indicates that our approach effectively learns the geometric features of objects, enabling accurate part segmentation. In interactive segmentation, our method also performs the best, thanks to our unique prompt point segmentation head and IOU prediction module. These components enable precise multi-scale segmentation predictions and automatically select the optimal results.

The comparisons on PartObj-Tiny-WT and PartNetE are shown in Table 3. Since watertight data and point clouds lack connectivity information, PartField's performance is not as good as on non-watertight data. This again validates that our method effectively learns the geometric features of objects. Here, SAMesh will get stuck when processing watertight meshes due to the high number of faces. Another observation is that the metrics for interactive segmentation are lower than those for full segmentation. This is because, in the evaluation of interactive segmentation, the method can only rely on a single prompt point, focusing more on the accuracy of individual mask segmentation. This further validates the precision of our approach. Comparisons across various datasets and tasks, involving different data forms such as non-watertight meshes, watertight meshes, and point clouds, demonstrate the remarkable effectiveness, robustness and generalization ability of our methods, confirming its superior performance under diverse conditions.

We also conduct a qualitative comparison of our method and previous methods on PartObj-Tiny for full segmentation with connectivity, as shown in Figure 4. SAMesh tends to over-segment objects, while other methods struggle with handling complex objects, resulting in inaccurate masks, failure to separate multiple part masks, and other segmentation errors. However, our method can accurately segment complex objects, even performing part segmentation on the lizard and beetle in the scene of the last row. The lower left corner of Figure 4 shows a comparison between our method and PartField on the PartObj-Tiny-WT dataset for full segmentation without connectivity. PartField struggles to segment watertight meshes, while our method can maintain the segmentation quality. The lower right corner of Figure 4 also illustrates the interactive segmentation results of our method and Point-SAM given the green point prompts, where our segmentation results exhibit accurate boundaries and scales. The results show that our method can predict masks with a reasonable number of categories, clear and accurate boundaries, and can handle complex models as well as different types of data and tasks. They also demonstrate the great generalization and robustness of our approach.

More segmentation results of our method on PartObj-Tiny, PartObj-Tiny-WT and AI-generated models are shown in Figures 20, 21 and 22.

## A.5 APPLICATIONS DETAILS

**Multi-Prompts Auto-Segmentation.** In this setting, we have a strong condition where each prompt corresponds to a specific part of the object. Therefore, instead of using the predicted IoU to evaluate mask quality, we only need to select one mask for each part such that all masks collectively cover the entire object as much as possible while minimizing overlap between the masks. Suppose the user selects $K$ point prompts indicating $K$ parts of an object. We directly predict 3 masks for each point prompt, resulting in $3K$ masks. For each prompt, we start from the smallest mask and progressively attempt to switch to slightly larger masks until the entire object is covered, while ensuring that the intersection between masks of different parts is minimized.

**Hierarchical Part Segmentation.** After segmentation, we collect the point-wise features $\mathbf{f}_p$ of the points corresponding to each part and compute the average feature of these points as the feature for each part. We then directly employ hierarchical clustering on these average features to obtain hierarchical segmentation results.

**Part Generation.** We employ HoloPart Yang et al. (2025a) to validate that our more accurate segmentation masks can significantly improve its performance. HoloPart requires segmented part point clouds as input and outputs complete components for each part. Originally, it used segmentation results from SAMPart3D; here, we replace those results with our own.

**Interactive Segmentation System.** We have built a visual and interactive segmentation system based on our P³-SAM. As shown in Figure 11, this system supports real-time segmentation by allowing users to select a prompt point. It also enables users to choose from three predicted masks and visualize the corresponding features.

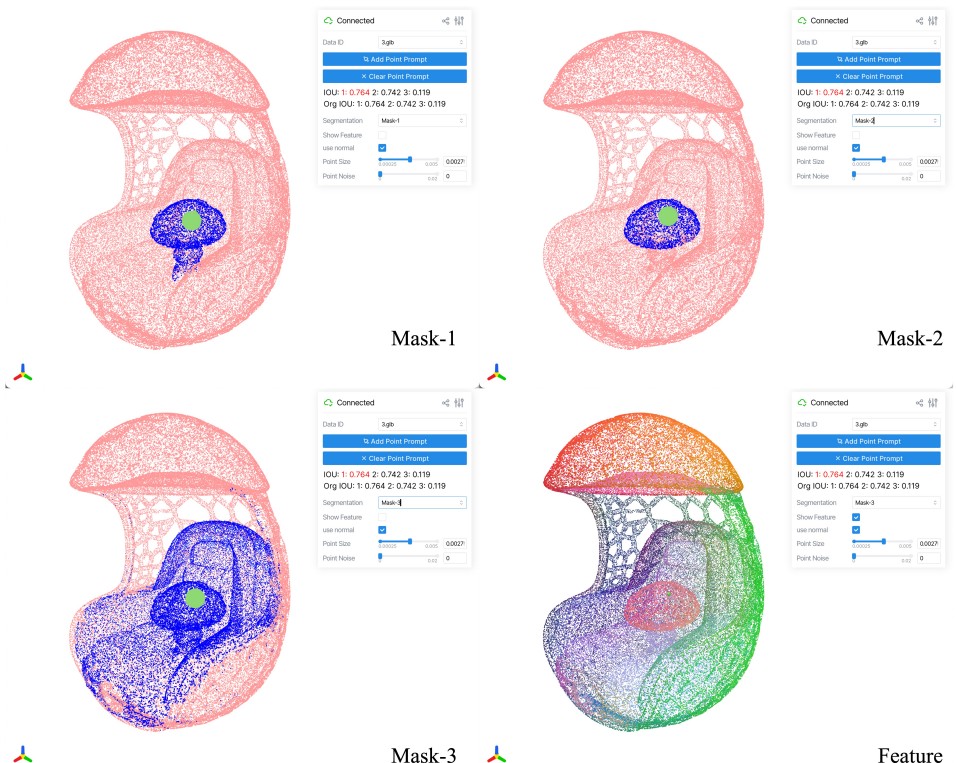

Figure 11: Our interactive segmentation system based on our P³-SAM.

## A.6    MORE EXPERIMENTS AND ANALYSIS

### A.6.1    ABLATION STUDIES ON WATERTIGHT DATA

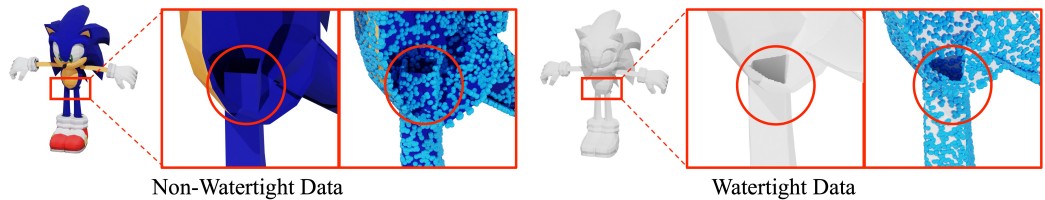

Figure 12: The comparison between non-watertight and watertight data on meshes and sampled point clouds.

Even though point clouds are relatively sparse representations, sampling from non-watertight meshes retains the characteristics of those meshes. As shown in Figure 12, we visually demonstrate the differences between watertight and non-watertight data. For example, in the human body model, the leg and body parts may intersect, leading to overlapping sampled point clouds. This results in discontinuous distributions of points and normals in these regions. In contrast, watertight meshes have connected leg and body parts, resulting in continuous distributions of points and normals. The discontinuities in point cloud and normal distributions provide more distinct features,

making it easier for networks to segment them without truly learning the semantic features of the object. Our experiments also show that segmentation metrics are lower for watertight data, confirming that segmenting watertight data is more challenging.

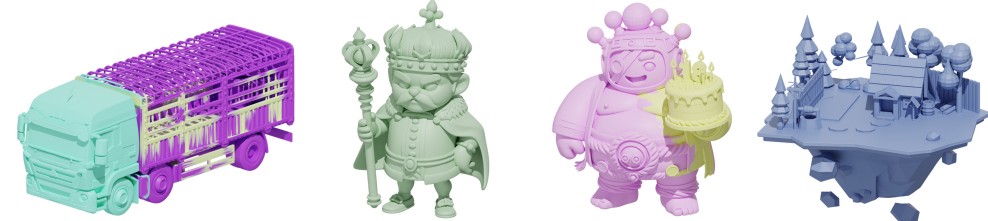

Figure 13: Our method, when trained only with non-watertight data, cannot segment watertight 3D generation models effectively.

We also conduct experiments on an ablated dataset without watertight data. We train our method on a dataset that do not include watertight data and evaluate it on 3D generated models. The results are present in Figure 13. Since these generated models are watertight, while this method is not trained on watertight data, the network fails to segment these models effectively, resulting in only one or two parts being identified. Therefore, the importance of watertight data in our method's performance is significant.

### A.6.2    ABLATION STUDIES ON DATASET SCALE

Table 5: The evaluation results of our method trained on various amounts of training data.

| Data Number | 100K | 500K | 1M | Full |
|---|---|---|---|---|
| IoU | 56.42 | 56.49 | 56.56 | **57.06** |

We conducted experiments to evaluate the impact of dataset scale on network performance by varying the amount of training data used. We train our model on datasets of 100K, 500K, 1M, and the full dataset ($\sim$2M). To ensure a fair comparison, all models are trained on 32 H20 GPUs with identical settings, including the same batch size, for 40K steps. We evaluate the "Full Segmentation without Connectivity" task on the PartObj-Tiny dataset. The evaluation results are shown in Table 5. It is evident that reducing the dataset size negatively impacts model performance. In addition to quantitative evaluations, we also conduct a qualitative analysis of the models' performance on 3D generated data, as shown in Figure 14. Reducing the dataset size significantly affects the model's generalization ability, leading to poorer segmentation results on 3D generated data. Note that since the full data training here only comprises 40K steps, the segmentation results in the figure are poorer compared to those presented in the paper.

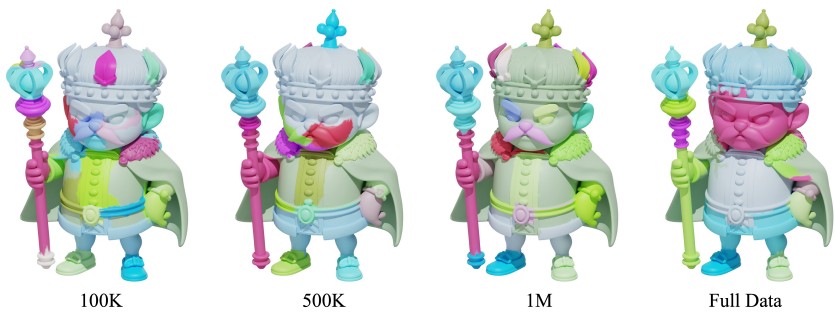

| 100K | 500K | 1M | Full Data |

Figure 14: The segmentation results of our method trained on various amounts of training data.

Table 6: The evaluation results of the compared methods trained on our dataset.

| | Seg. w/ Connect | | | Seg. w/o Connect | | | Inter. Seg. | |
|---|---|---|---|---|---|---|---|---|
| | PartField | Sonata+CL | Ours | PartField | Sonata+CL | Ours | PointSAM | Ours |
| Org. Data | 79.18 | - | - | 53.93 | - | - | 27.91 | - |
| Ours Data | 78.33 | 76.41 | **81.14** | 55.62 | 50.02 | **59.88** | 28.64 | **51.23** |

### A.6.3 COMPARISON ON LARGE-SCALE DATASET

To validate the effectiveness and innovativeness of our approach, we train different methods on our full dataset and evaluate their performance, including PartFieldLiu et al. (2025), PointSAMZhou et al. (2024), and an ablated method . The ablated method "Sonata+CL" combines feature extraction from SonataWu et al. (2025) with the contrastive learning loss from PartField. The ablated method "Sonata+CL" differs from PartField primarily in its feature extraction module, which is replaced with the feature extraction module from Sonata. Compared to our full method, "Sonata+CL" lacks the specialized segmentation head that we introduced. The evaluation results on various tasks of the PartObj-Tiny dataset are shown in Table6. Even when the comparison methods are trained on our full dataset, their performance do not surpass those of our method, demonstrating the effectiveness and innovativeness of our approach. Notably, PartField's performance on the "Full Segmentation with Connectivity" task show a slight decline, further highlighting the stability of our method when trained on large-scale data. The performance of "Sonata+CL" do not surpass that of PartField or our method, further demonstrating the uniqueness and innovativeness of our approach, particularly our two-stage segmentation module. The superiority of our method stems not only from the quality and scale of the data but also from our unique and innovative network design.

### A.6.4 ABLATION STUDIES ON HYPERPARAMETERS OF AUTOMATIC SEGMENTATION

Table 7: The evaluation results of automatic segmentation with various NMS thresholds.

| NMS Threshold | 0.70 | 0.80 | 0.85 | **0.90** | 0.95 |
|---|---|---|---|---|---|
| IoU | 59.90 | 59.94 | 59.87 | 59.88 | 59.55 |

Table 8: The evaluation results and running times of automatic segmentation with various prompt numbers.

| Prompt Numbers | 10 | 50 | 100 | 200 | **400** | 600 | 800 |
|---|---|---|---|---|---|---|---|
| IoU | 44.83 | 56.44 | 57.73 | 58.96 | 59.88 | 59.90 | 60.31 |
| Time (Second) | 0.581 | 1.569 | 2.416 | 4.203 | 8.214 | 13.308 | 17.016 |

Our automatic segmentation involves two hyperparameters: the number of prompt points and the overlap threshold used in the NMS process to determine whether two masks overlap. During the automatic segmentation process, the network "self-determines" the granularity of objects without human intervention, similar to how NMS determines the number of objects in image object detection. To verify the impact of these two hyperparameters on automatic segmentation, we conduct ablation studies on these two hyperparameters evaluated on the "Full Segmentation without Connectivity" task of the PartObj-Tiny dataset. The evaluation results are shown in Tabels 7 and 8. The default values for these two hyperparameters are 0.9 for the NMS overlap threshold and 400 for the number of prompt points. It is evident that the method's performance is not sensitive to the NMS threshold. This indicates the accuracy of our approach, meaning that the segmentation masks for the same part under different numbers of prompt points are quite similar, and the masks of different parts rarely overlap. Even with smaller thresholds, there is no significant risk of incorrectly determining whether two masks overlap. As the number of prompt points increases, the performance of our method gradually improves. This is because a higher number of prompt points can better cover all parts, allowing the method to segment as many parts as possible. However, this also leads to an increase in computational time. Therefore, we opt for a balanced setting of 400 prompt points, which

provides a good trade-off between performance and efficiency. We also demonstrate the impact of different numbers of prompt points on segmentation quality in Figure 15. This figure visually illustrates that as the number of prompt points increases, the number of parts identified and segmented also increases, with more smaller parts being successfully segmented out.

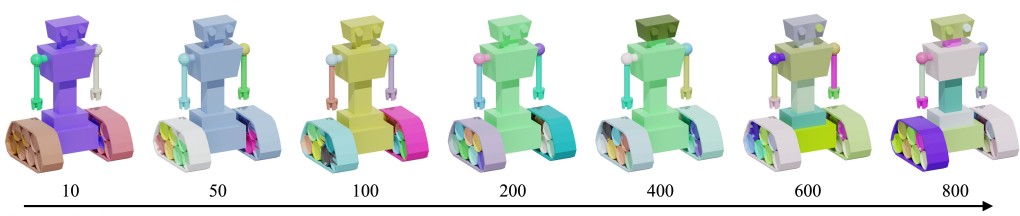

Prompt Number

Figure 15: The segmentation results and running times of automatic segmentation with various prompt numbers.

A.6.5    ABLATION STUDY ON NOISY POINTS AND NORMALS

Table 9: The evaluation results of our methods with varying levels of noise and with normals removed.

| Noise $s_{max}$ | 0 | 0.001 | 0.005 | 0.01 | 0.05 | 0.1 |
|---|---|---|---|---|---|---|
| w/ Normal | **59.88** | 59.34 | 58.41 | 57.34 | 31.75 | 21.13 |
| w/o Normal | 57.56 | 57.22 | 56.04 | 55.02 | 29.25 | 20.70 |

Our method incorporates data augmentation techniques, including randomly removing normals and adding noise to point clouds and normals. This enhances our method's robustness in handling scenarios where points and normals are missing or noisy. To validate this, we conduct experiments on the "Full Segmentation without Connectivity" task using the PartObj-Tiny dataset. Following the augmentation methods detailed in Section A.3.2, we add random noise to the point clouds and normals and also remove the normals entirely. The evaluation results are shown in Table 9. The results indicate that our method maintains SOTA performance when the random noise is less than 0.01. Additionally, the performance does not significantly degrade in the absence of normals. These results demonstrate the excellent robustness of our approach.

A.6.6    EVALUATION ON MORE DATASET

We compare PartFieldLiu et al. (2025) and our method on the car and airplane categories in the 3DCOMPAT++ datasetSlim et al. (2023); Li et al. (2022b) (total 68 meshes), including both coarse and fine granularities, to supplement our evaluation. 3DCoMPaT++ is a multimodal dataset for compositional 3D research with detailed part-level annotations. It covers 41 shape categories, 275 fine-grained part categories, and 293 fine-grained material classes that can be compositionally applied to parts of 3D objects. The evluation results are shown in Tabel 10. The labels in the 3DCOMPAT++ dataset are at the semantic level, meaning that two parts with the same semantic meaning are grouped under the same mask. This differs from instance-level labeling and thus may result in lower metrics. However, the metrics on this dataset can still serve as a valid measure of model segmentation performance. As shown in Table 10, our method outperforms PartField in both coarse and fine granularities, further demonstrating the superiority of our approach.

A.6.7    EVALUATION ON REAL-WORLD DATA

We evaluate our method on real-world data from OmniObject3DWu et al. (2023), as shown in the Figure 16. We demonstrate the segmentation results of our method on categories including chairs, dolls, dinosaurs and motorcycles. These datasets are real scans, with point clouds and normal distributions that differ from our training data and contain noise. Our method can still segment these

Table 10: The comparison of our method and PartField on 3DCOMPAT++.

| IoU | Airplane | Car | AVG. |
|---|---|---|---|
| PartField(Coarse) | 38.09 | 33.01 | 35.54 |
| Ours(Coarse) | **40.23** | **36.24** | **38.23** |
| PartField(Fine) | **32.65** | 20.67 | 26.66 |
| Ours(Fine) | 16.83 | **38.07** | **27.45** |

models, showcasing its generalization ability and robustness on real-world data. However, our method has failure cases on real-world data, as shown in Figure17. On overly smooth objects, our method fails to segment parts, while for objects with significant noise, our method results in over-segmentation.

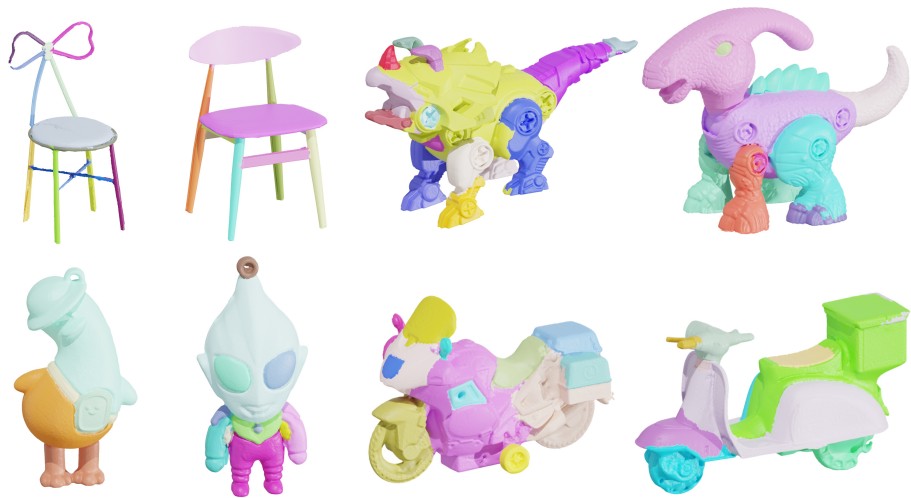

Figure 16: The segmentation results of our method on real scanned data of various categories including chairs, dolls, dinosaurs and motorcycles.

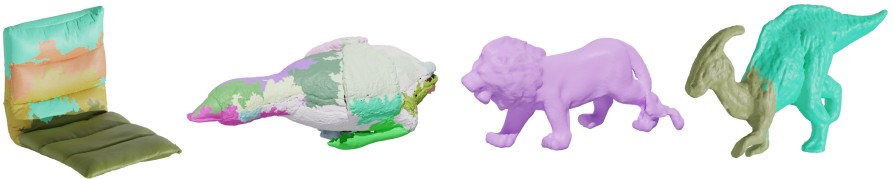

Figure 17: The failure cases of our method on real scanned data.

### A.6.8 CROSS-SHAPE CONSISTENCY

We follow the method described in Section 4.3 of PartField's paper Liu et al. (2025) to evaluate cross-shape consistency on more complex 3D generated data. The visualization results as shown in Figure18. We conduct cross-shape consistency evaluations on two human shape models and one dinosaur model. On the human shape models, both our method and PartField show similar consistency. However, when evaluating cross-shape consistency between the human shape and dinosaur models, our method outperforms PartField. Specifically, our method results in consistent colors for the head and right hand of both the human shape and dinosaur models, whereas PartField does not achieve this consistency. This indicates that our method can exhibit cross-shape consistency across different categories, demonstrating its excellent generalization ability and capability to understand

model semantics. Here, we simplified the number of faces in the 3D generated models to ensure that the algorithm for measuring cross-shape consistency could run properly and efficiently. We also demonstrate feature visualizations for different objects using PCA to reduce the dimensionality of each face's feature. Our method can extract more detailed features compared to PartField. This further demonstrates that our model can not only extract high-level semantics but also retain local geometric detail features, highlighting the superiority of our approach.

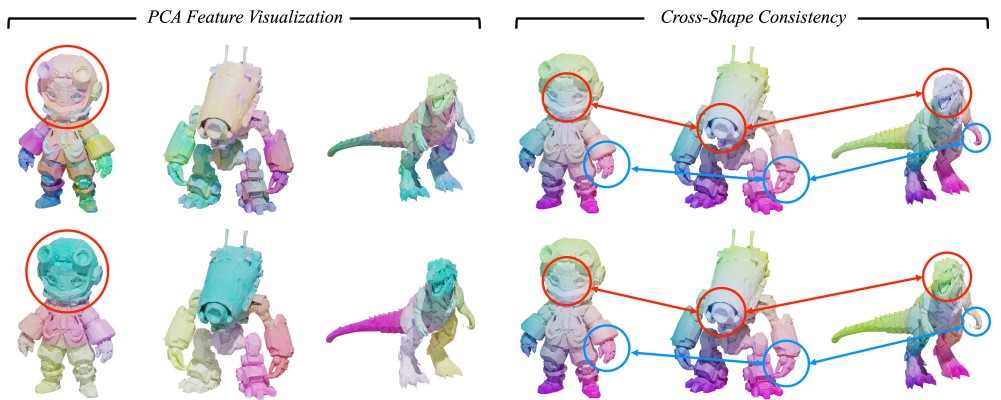

Figure 18: The PCA feature visualization and cross-shape consistency of our method and PartField.

### A.6.9 SEGMENTATION OF INTERNAL STRUCTURES

We evaluate our method on objects with internal structures, as shown in Figure 19. Our method successfully segments the interiors of cars, including components such as seats and steering wheels. As long as internal structures have surface point clouds, our method can perform segmentation.

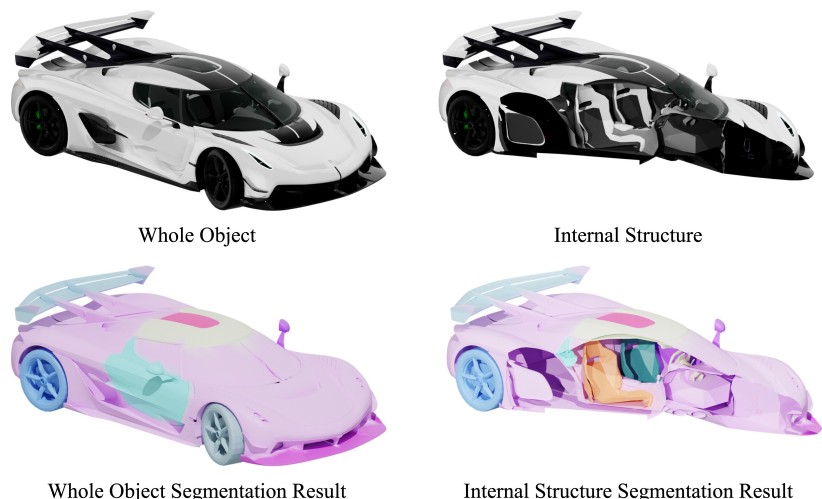

Figure 19: The internal structure segmentation result of our method.

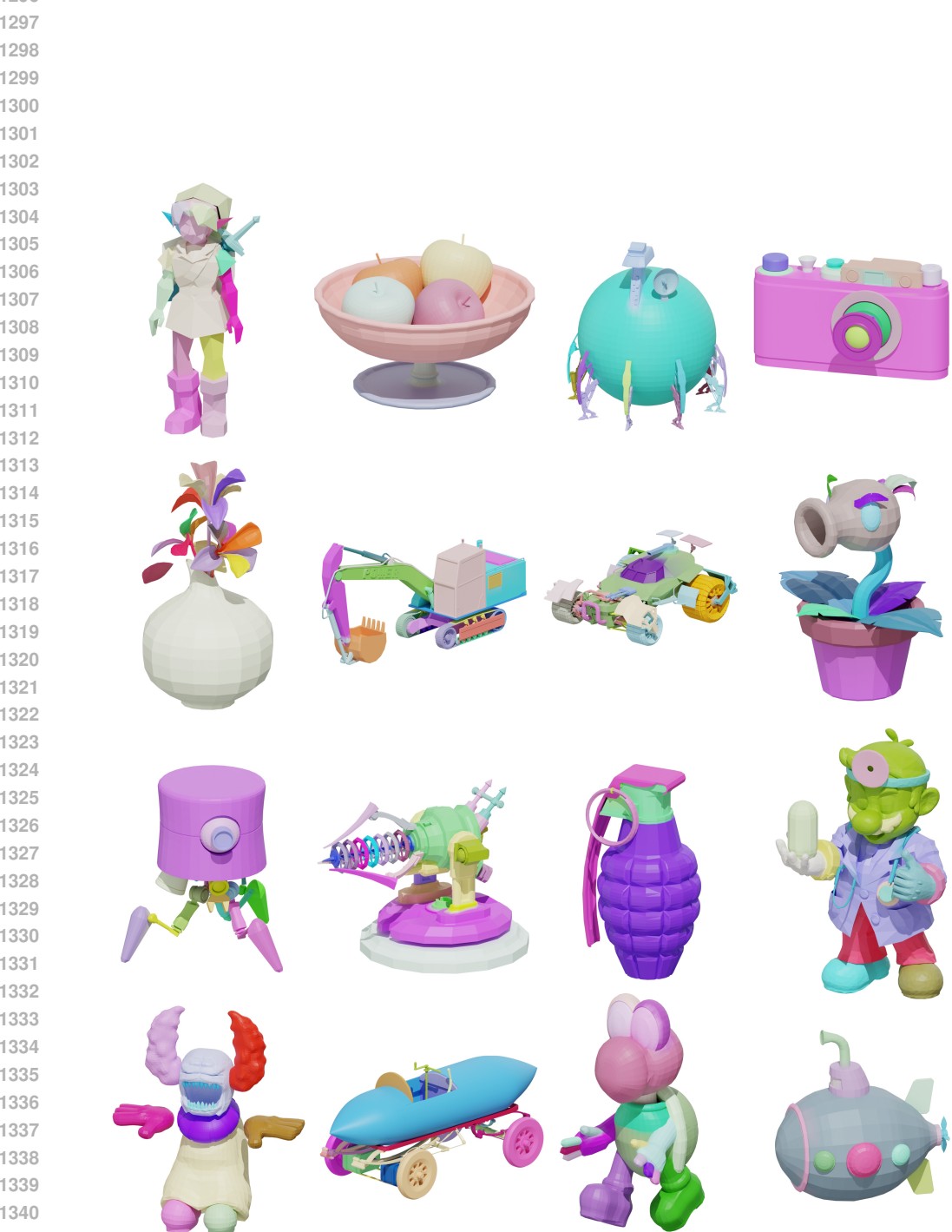

Figure 20: More results on PartObj-Tiny.

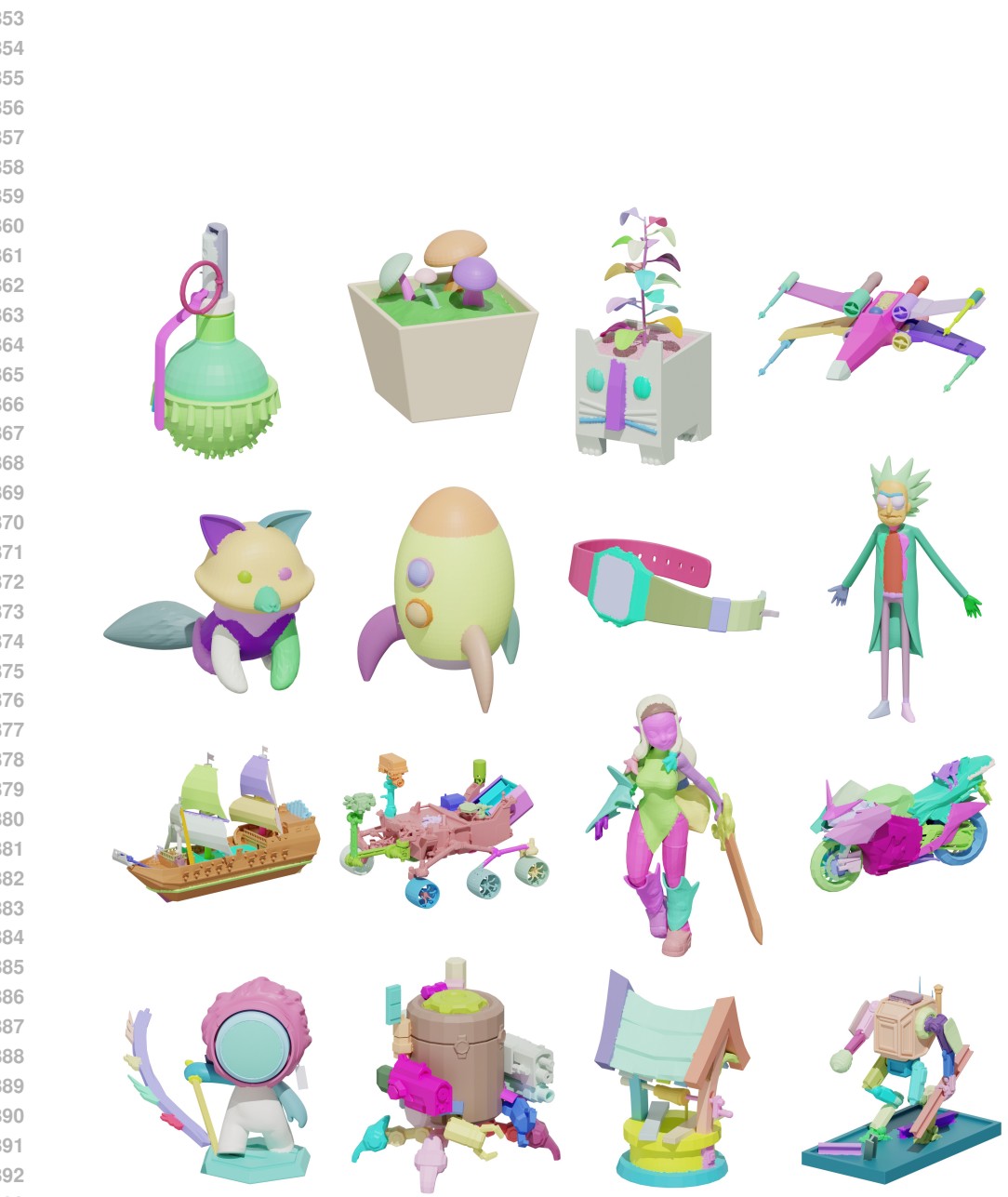

Figure 21: More results on PartObj-Tiny-WT.

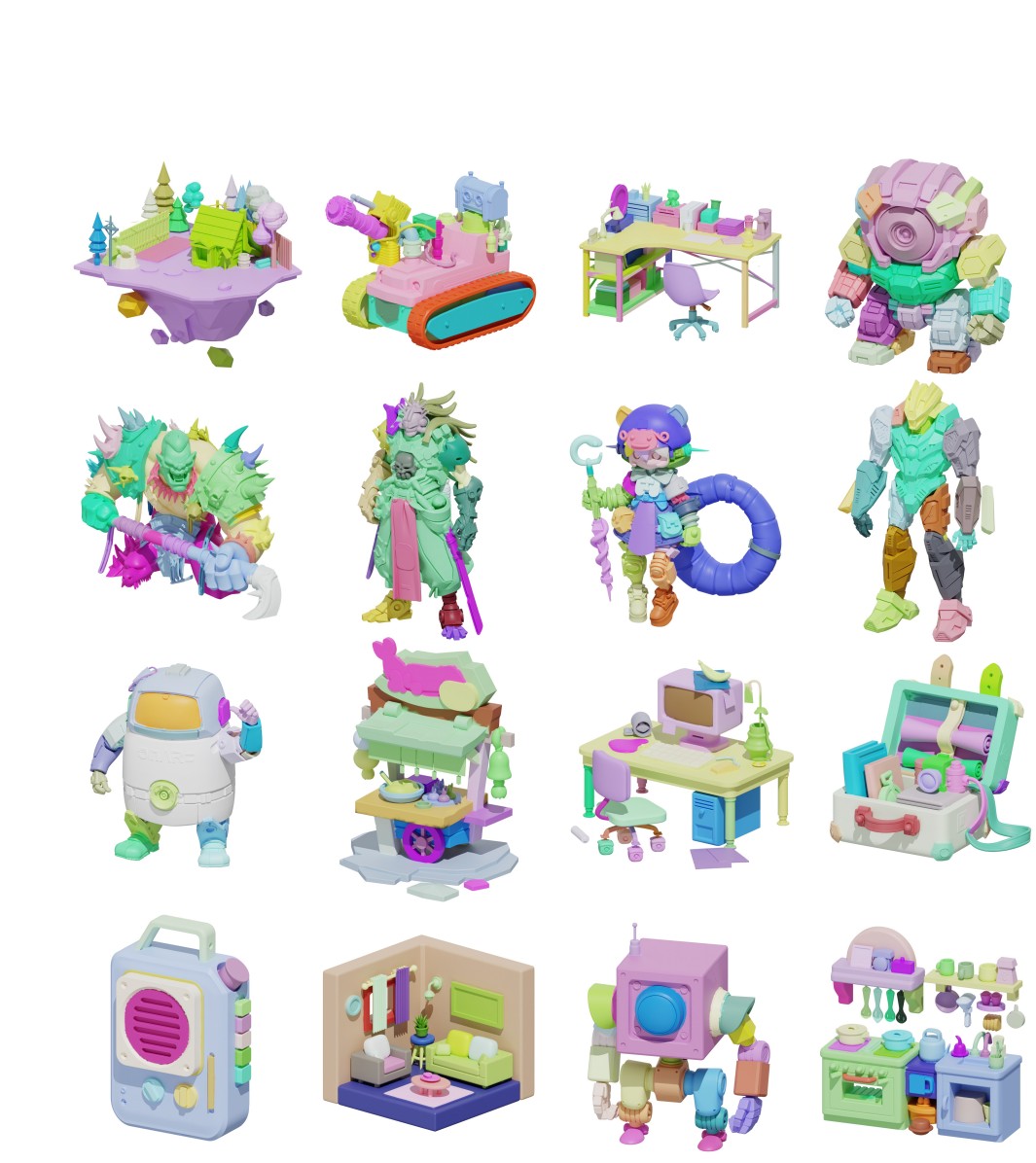

Figure 22: More results on AI generated models.

