# OpenReview forum: "P$^3$-SAM: Native 3D Part Segmentation"
_ICLR.cc/2026/Conference — Submitted to ICLR 2026_

### Official Review · Reviewer_UkFr · 2025-10-17

**Soundness:** 3
**Presentation:** 3
**Contribution:** 2
**Rating:** 4
**Confidence:** 4

**Summary:**

This paper proposes a native 3D point-promptable part segmentation model named P3-SAM, aiming to achieve fully automatic and precise segmentation of arbitrary 3D objects. The core contribution lies in constructing a native 3D part segmentation dataset containing nearly 3.7 million models, based on which a segmentation model with excellent performance was trained. Experimental results show that this method surpasses existing methods on multiple benchmarks. Nevertheless, there is still room for improvement in the paper regarding the innovativeness of its methodology, the details of its data processing, and the depth of its experiments.

**Strengths:**

1.  **Pioneering Native 3D Dataset:** The core contribution is a massive native 3D part segmentation dataset (nearly 3.7 million models), which directly resolves the 2D-3D data gap and avoids issues from 2D-lifting.

2.  **State-of-the-Art Performance and Robustness:** The model demonstrates convincing state-of-the-art performance across multiple benchmarks and shows exceptional robustness on diverse data modalities, including non-watertight meshes, watertight meshes, and point clouds.

3.  **Complete Automatic Segmentation Pipeline:** The paper introduces a full end-to-end automatic segmentation pipeline, using Farthest Point Sampling (FPS) for prompt generation and Non-Maximum Suppression (NMS) for mask filtering, enabling segmentation without human intervention.

**Weaknesses:**

1.  **Limited Methodological Innovation:** The paper's state-of-the-art performance is largely attributed to its large-scale, novel 3D dataset rather than breakthroughs in the model architecture itself. The model appears to be more of a strong engineering implementation than a novel theoretical contribution, which may cast doubt on the superiority of its design.

2.  **Restricted Model Interactivity:** The model's support for only single-point prompts limits its application in complex scenarios where objects might require multiple points or bounding boxes for precise definition. The paper fails to justify the exclusion of other interaction methods or discuss how the model handles ambiguous single-point prompts.

**Questions:**

1.  **Questions on Model Generalization:** The model is primarily trained on synthetic 3D data; its generalization performance on real-world 3D scanned data is unclear.

2.  **Request for Specific Limitation Examples:** The paper states the model lacks an understanding of spatial volume. This raises the question of whether it would fail to segment real-world point clouds with complex internal structures or closely adjacent parts. Visual failure cases are requested to illustrate this limitation.

**Details Of Ethics Concerns:**

See the Questions section for details.

---

> ### Author Response · Authors · 2025-11-20
>
> Thank you for your thoughtful and constructive suggestions, which have significantly shaped our improvements and are highly valued by our team.
> Here are our responses:
>
> > **W1:** Limited methodological innovation.
>
> We did draw inspiration primarily from the concept of prompt-based segmentation in SAM, but instead of mechanically replicating it, we thoroughly integrated this idea with the characteristics of point clouds and the functionalities required for automatic full segmentation tasks to design our network. For instance:
> - Feature Extractor: We adopted a SOTA network architecture and pretrained weights that are specially designed for point clouds,
> - Prompt: We discarded the complex prompt design and prompt training, and instead adopted a single prompt point to focus on 3D object full segmentation.
> - Mask Decoder: We developed a unique two-stage segmentation head that only uses an MLP, without employing the Two Way Transformer as in SAM.
> - Data: There is no data engine in our method. We directly train our model on our large-scale preprocessed native 3D data.
>
> These are all distinctive and novel designs of ours.
>
> To further illustrate the advantages of our method's architecture, we trained different methods on our full dataset and evaluated their performance.Specifically, we trained PartField, PointSAM, and an ablated method to validate the effectiveness and innovativeness of our approach. The ablated method "Sonata+CL" combines feature extraction from Sonata with the contrastive learning loss from PartField. The ablated method "Sonata+CL" differs from PartField primarily in its feature extraction module, which has been replaced with the feature extraction mechanism from Sonata. Compared to our full method, "Sonata+CL" lacks the specialized segmentation head that we introduced.  The comparison results are presented in the tables below.
>
> | Full Seg. w/ C.| Org. Data| Our Data |
> | ---- |  ---- |  ---- |
> | PartField | 79.18 | 78.33 |
> | Sonata+CL | - | 76.41 |
> | Ours | - | 81.14 |
>
> | Full Seg. w/o C.| Org. Data| Our Data |
> | ---- |  ---- |  ---- |
> | PartField | 53.93 | 55.62 |
> | Sonata+CL | - | 50.02 |
> | Ours | - | 59.88 |
>
> | Inter. Seg. | Org. Data| Our Data |
> | ---- |  ---- |  ---- |
> | PointSAM | 27.91 | 28.64 |
> | Ours | - | 51.23 |
>
> Even when the comparison methods were trained on our full dataset, their metrics did not surpass those of our method, demonstrating the effectiveness and innovativeness of our approach. Notably, PartField's performance on the "Full Segmentation with Connectivity" task showed a slight decline, further highlighting the stability of our method when trained on large-scale data.
> The performance of "Sonata+CL" did not surpass that of PartField or our method, further demonstrating the uniqueness and innovativeness of our approach, particularly our two-stage segmentation module.
> Thus, the superiority of our method stems not only from the quality and scale of the data but also from our unique and innovative network design.
>
>
> > **W2:** Restricted model interactivity.
>
> Methods such as multiple prompt points, positive and negative prompt points, bounding boxes, and other interactive methods are not unreasonable. However, in our goal of achieving automatic full segmentation, these methods do not provide significant benefits, so we did not implement them. First, other interactive methods would increase the training process and complexity, making it harder for the network to converge. Second, these methods are too complex, complicating the design of automatic post-processing algorithms. Third, a single prompt point segmentation scheme already achieves sufficiently accurate masks. Therefore, we only implemented single prompt point segmentation.
>
> We used three segmentation heads to address ambiguous single prompt point scenarios. The network predicts three masks at different scales and uses an IoU prediction module to estimate the IoU for each mask. Finally, the user can select one of the three masks or choose the mask with the highest IoU as the final segmentation result. By employing multiple segmentation heads responsible for masks at different scales, we can resolve ambiguities in both training and inference when using a single prompt point.

---

> > ### Author Response · Authors · 2025-11-20
> >
> > > **Q1:**  Generalization on real-world data.
> >
> > As suggested, we evaluated our method on real-world data from OmniObject3D [3], as shown in Figure 16 of the appendix. We demonstrated the segmentation results of our method on categories including chairs, dolls, dinosaurs and motorcycles. These datasets are real scans, with point clouds and normal distributions that differ from our training data and contain noise. However, our method can still segment these models, showcasing its generalization ability and robustness on real-world data.
> >
> > [3] OmniObject3D: Large-Vocabulary 3D Object Dataset for Realistic Perception, Reconstruction and Generation
> >
> > > **Q2:** Limitation examples
> >
> > Since our method is trained and performs inference on surface point clouds, it cannot segment space (i.e., any arbitrary point in space) like methods such as OctNet [1]. Therefore, we acknowledge that our network lacks an understanding of spatial volume.
> >
> > In Q1, we demonstrated the segmentation results of our method on real-world datasets. However, real data is typically acquired using 3D scanners, which, due to their operational principles, often fail to capture internal structures of objects. Therefore, we used synthetic data to test the performance of our method on data with complex internal structures. The segmentation results can be found in the Figure 19 of our appendix, where our method successfully segments the interiors of cars, including components such as seats and steering wheels.
> > As long as internal structures have surface point clouds, our method can perform segmentation.
> >
> > Our method has failure cases on real-world data, which we have already demonstrated in Figure 17 of the appendix. On overly smooth objects, our method fails to segment parts, while for objects with significant noise, our method results in over-segmentation.
> >
> > [1] OctNet: Learning Deep 3D Representations at High Resolutions

---

> ### Author Response · Authors · 2025-11-27
> **Rebuttal Feedback Request**
>
> Thank you for your insightful comments and dedication to evaluating our manuscript. We have submitted a detailed rebuttal addressing all your points. We kindly request your feedback on our responses at your earliest convenience, and we would be pleased to discuss any points further with you. Your guidance remains invaluable as we refine our work, and we sincerely appreciate your time and expertise.

---

### Official Review · Reviewer_cf8P · 2025-10-31

**Soundness:** 3
**Presentation:** 3
**Contribution:** 3
**Rating:** 8
**Confidence:** 4

**Summary:**

This paper introduces $P^{3}$-SAM, a 3D point-promptable part segmentation model that is natively trained on 3D data. It is a solid contribution to the field by moving away from 2D-lifting or 2D data engine approaches and building a truly native 3D solution. It also introduces a large-scale 3D part dataset of 3.7M models, which is a significant contribution to the community.

**Strengths:**

1. The curation of the 3.7M 3D model dataset with native 3D part annotations is a significant contribution in terms of number. The data curation details in Appendix A.3.1 are reasonable, and the handling of non-watertight and watertight data sounds novel to me.
2. The motivation to build a 3D native solution is meaningful and reasonable. It addresses real limitations in existing 2D-lifting and 2D data engine methods, such as 2D-3D data gaps, 3D consistency issues, and computational overhead from multi-view rendering.
3. $P^{3}$-SAM outperforms the SOTA methods like PartField in the benchmark.

**Weaknesses:**

1. PartObj-Tiny contains only 200 shapes across 8 categories, which is insufficient to demonstrate the performance of a model convincingly. Although there is another evaluation on PartNetE dataset having 1,906 shapes, evaluating on larger datasets like 3DCOMPAT++ [1] will make the claims much stronger.


[1] 3DCOMPAT++: A comprehensive dataset for 3D object understanding with fine-grained part annotations

**Questions:**

1. Given weakness 1, would it be possible to evaluate the performance on the car and airplane categories (68 meshes in total) from 3DCOMPAT++ [1] and compare the results with PartField? It will also be great if the evaluation could be performed on both fine-grained parts and coarse parts.
2. PartField shows some emergent cross-shape consistency when exploring the features. $P^{3}$-SAM is trained on a larger (maybe better) dataset, could you also give some qualitative and quantitative (if possible) comparison on a small scale with PartField?

[1] 3DCOMPAT++: A comprehensive dataset for 3D object understanding with fine-grained part annotations

---

> ### Author Response · Authors · 2025-11-20
>
> We sincerely thank you for your positive acknowledgment of our work and for providing such valuable insights, which have been instrumental in guiding our revisions.
> Here are our responses:
>
> > **W1&Q1:** More evaluation on 3DCOMPAT++.
>
> Currently, we have conducted comparative evaluation on the 3DCOMPAT++ dataset.
> As suggested, we compared PartField and our method on the car and airplane categories in the 3DCOMPAT++ dataset, including both coarse and fine granularities. The results are presented in the table below.
>
> |IoU|	Airplane|	Car|	AVG.|
> | ---- | ---- | ---- | ---- |
> |PartField(Coarse)|	38.09|	33.01|	35.54|
> |Ours(Coarse)|	40.23|	36.24|	38.23|
> |PartField(Fine)|	32.65|	20.67|	26.66|
> |Ours(Fine)|	16.83|	38.07|	27.45|
>
> The labels in the 3DCOMPAT++ dataset are at the semantic level, meaning that two parts with the same semantic meaning are grouped under the same mask. This differs from instance-level labeling and thus may result in lower metrics. However, the metrics on this dataset can still serve as a valid measure of model segmentation performance. As shown, our method outperforms PartField in both coarse and fine granularities, further demonstrating the superiority of our approach.
> We have added this supplementary evaluation to our paper.
>
> > **Q2:** Cross-shape consistency.
>
> We followed the method described in Section 4.3 of PartField's paper to evaluate cross-shape consistency on more complex 3D generated data. The visualization results can be found in Figure 18 of the appendix. We conducted cross-shape consistency evaluations on two human shape models and one dinosaur model. On the human shape models, both our method and PartField showed similar consistency. However, when evaluating cross-shape consistency between the human shape and dinosaur models, our method outperformed PartField. Specifically, our method resulted in consistent colors for the head and right hand of both the human shape and dinosaur models, whereas PartField did not achieve this consistency. This indicates that our method can exhibit cross-shape consistency across different categories, demonstrating its excellent generalization ability and capability to understand model semantics.
> Here, we simplified the number of faces in the 3D generated models to ensure that the algorithm for measuring cross-shape consistency could run properly.

---

> ### Author Response · Authors · 2025-11-27
> **Rebuttal Feedback Request**
>
> Thank you for your insightful comments and dedication to evaluating our manuscript. We have submitted a detailed rebuttal addressing all your points. We kindly request your feedback on our responses at your earliest convenience, and we would be pleased to discuss any points further with you. Your guidance remains invaluable as we refine our work, and we sincerely appreciate your time and expertise.

---

### Official Review · Reviewer_CnDT · 2025-11-02

**Soundness:** 3
**Presentation:** 3
**Contribution:** 3
**Rating:** 6
**Confidence:** 4

**Summary:**

This paper proposes a 3D point feature based segmentation method that leverages the speed and high quality segmentation. It uses a single prompt point, two stage segmentor and IoU classifier to automatically and accurately segment complex 3D objects into their constituent parts. The first stage produces multi-scale mask predictions, while the second stage refines them using global contextual features. The IoU predictor then evaluates and selects the best mask, enabling robust and fully automated part segmentation.

**Strengths:**

- Native 3D formulation without lifting based on 3D point-based feature, avoid multi -view rendering or 2D-to-3D lifting which reduced the domain gasps and improved the robustness.
- Large-scale and high-quality 3D dataset with automatic part-level annotations. It is a strong generalization across watertight and non-watertight  objects.
- Efficient two-stage segmentation as mentioned in paper only 8s for a full and interactive segmentation.

**Weaknesses:**

- High probability of removing normals while training (0.3)? With the probability of 0.3 it may remove normals during data augmentation and improves robustness to missing information. But it could weaken the model’s geometric precision.
- Lack of volumetric and semantic understanding because the training data does not contain labels and might have weak boundary prediction. The dataset was created using automatic geometric merging based on area thresholds and adjacency rules. This may merge distinct but small parts and remove fine structural details.
- The automatically generated segmentation using geometric heuristics may introduce bias and remove some fine-grained parts which are hard to apply on real world data.  The dataset has removed the imbalance parts, it might be hard to apply on moderately complex and regular structure.

**Questions:**

- How do you validate the created dataset, since the algorithm combines small parts and merges adjacent meshes automatically? Was the manual inspection involved to ensure that the resulting part annotations remain accurate and do not over-merge distinct components?

---

> ### Author Response · Authors · 2025-11-20
>
> We are deeply grateful for your recognition of our contributions and for your constructive feedback, which we consider essential for elevating the quality of our manuscript.
> Here are our responses:
>
> > **W1:** High probability of removing normals.
>
> To validate the robustness of our method on normals, we conducted experiments on the "Full Segmentation without Connectivity" task using the PartObj-Tiny dataset. Following the augmentation methods detailed in the appendix, we added random noise to the point clouds and normals and also removed the normals entirely. The results are presented in the table below.
>
> |Noise $s_{max}$|	0|	0.001|	0.005|	0.01|	0.05|	0.1|
> | ---- | ---- | ---- | ---- | ---- | ---- | ---- |
> |w/ Normal|	59.88|	59.34|	58.41|	57.34|	31.75|	21.13|
> |w/o Normal|	57.56|	57.22|	56.04|	55.02|	29.25|	20.70|
>
> Removing normals leads to a loss in the accuracy of our method; however, the performance does not significantly degrade, and our approach remains at the leading level. These results demonstrate the excellent robustness of our approach.
>
> > **W2:** Lack of volumetric and semantic understanding
>
> The lack of volumetric understanding has almost no impact on boundary detection since we train and infer on surface point clouds, where the data is aligned. However, it is true that our method cannot segment spaces, i.e., determine which part an arbitrary position in space belongs to.
>
> The lack of semantic data could affect the quality of mask segmentation. On one hand, data with semantic labels tends to have better mask quality because each part consistently corresponds to a specific semantic meaning. On the other hand, semantic labels allow the network to learn the relationship between part features and semantics, thereby encouraging the network to generate masks that align with semantic meanings. Therefore, using data with semantic labels is highly likely to improve network performance. However, existing labeled datasets such as PartNeXt[1] and PartNet[2] suffer from limited categories (approximately 50 categories) and small dataset sizes (around 40,000 data points). Training on these datasets makes it difficult to generalize to arbitrary objects. In our future work, we will address this issue by constructing large-scale datasets with semantic part annotations
>
> We acknowledge that merging small parts introduces some low-quality noisy data, where the segmentation masks are inaccurate, which could impact the model. However, we believe that the sheer volume of data can mitigate this issue, allowing the network to learn an "average" correct segmentation from the large dataset, which is one of the primary benefits of having extensive data.
>
> [1] PartNeXt: A Next-Generation Dataset for Fine-Grained and Hierarchical 3D Part Understanding
>
> [2] PartNet: A Large-scale Benchmark for Fine-grained and Hierarchical Part-level 3D Object Understanding
>
> > **W3:** Hard to apply on real-world data
>
> Our method has not been trained on extremely complex data because we filtered out datasets containing too many parts and imbalanced data. Therefore, it is difficult to directly segment extremely complex objects.However, with simple modifications, our method can achieve fine-grained partitioning. For instance:
> - During mask prediction, instead of selecting the mask with the highest IoU, we could output the smallest mask among the predictions. This approach would lead to finer granularity in the final segmentation results.
> - Or we could recursively apply our method to subdivide parts. Specifically, after initially identifying a part, we can reapply our segmentation algorithm to that part to further divide it into smaller components.
>
> We also evaluated our method on real-world data from OmniObject3D [3], as shown in Figure 16 in the appendix. Our method can segment real scanned data, and although the geometry of these real-world datasets is not very complex, it still demonstrates the robustness of our approach on real data.
>
> [3] OmniObject3D: Large-Vocabulary 3D Object Dataset for Realistic Perception, Reconstruction and Generation

---

> > ### Author Response · Authors · 2025-11-20
> >
> > > **Q1**: Dataset curation.
> >
> > The construction of our dataset adheres to a few simple principles and filtering criteria mentioned in the paper, and it does not involve any manual intervention, significantly reducing human labor. We acknowledge that this approach introduces some low-quality noisy data, where the segmentation masks are inaccurate, which could impact the model. However, we believe that the sheer volume of data can mitigate this issue, allowing the network to learn an "average" correct segmentation from the large dataset, which is one of the primary benefits of having extensive data.
> >
> > The automatic merging strategy is designed to ignore the impact of extremely small components on the overall segmentation. These tiny components often appear in highly complex models designed by professionals. Such components typically occupy very small masks and thus have minimal impact on prediction accuracy. For example, components like rivets on large machinery occupy a very small proportion of the entire object. When the object is sampled as a point cloud, these components are rarely captured or are represented by only a few points.
> > Specific data curation details can be found in the appendix. Among these, we adopted a very conservative strategy during merging, combining only parts that occupy less than 1% of the area. This approach prevents the excessive merging of different components, which could otherwise lead to a large number of unreasonable results.

---

> ### Author Response · Authors · 2025-11-27
> **Rebuttal Feedback Request**
>
> Thank you for your insightful comments and dedication to evaluating our manuscript. We have submitted a detailed rebuttal addressing all your points. We kindly request your feedback on our responses at your earliest convenience, and we would be pleased to discuss any points further with you. Your guidance remains invaluable as we refine our work, and we sincerely appreciate your time and expertise.

---

### Official Review · Reviewer_imJU · 2025-11-03

**Soundness:** 3
**Presentation:** 3
**Contribution:** 3
**Rating:** 4
**Confidence:** 4

**Summary:**

This paper proposes a novel method for 3D part segmentation by training a 3D-native part segmentation model. The model follows the paradigm of the 2D SAM, taking a 3D point as an input prompt and outputting three masks along with an IoU prediction. It is trained on 3.7 million 3D models with reasonably accurate segmentation labels.

**Strengths:**

1. The paper addresses an important task and is well-motivated and technically sound.

2. It presents a method for cleaning a large-scale 3D dataset and obtaining part annotations. The proposed model is one of the first to be trained with such large-scale 3D native part supervision.

3. The paper is well-written and easy to follow.

4. The proposed method appears to outperform previous baselines across multiple setups.

5. The paper provides ablation studies with respect to the network architecture.

**Weaknesses:**

1. **Method novelty and comparison with existing work.**
   From a methodological perspective, the high-level idea of the proposed model and training scheme closely follows that of 2D SAM, and thus lacks substantial novelty or conceptual breakthrough. Moreover, similar attempts to extend 2D SAM to 3D already exist (e.g., *PointSAM*). The paper should include a more detailed discussion comparing its network design and training scheme to 2D SAM and PointSAM, highlighting the specific differences, motivations, and potential advantages.

2. **Dataset curation and its scientific value.**
   One of the main contributions of this paper appears to be the curation of a large-scale part-annotated 3D dataset by leveraging the existing connectivity and grouping information embedded in raw 3D meshes. While the authors briefly describe the process of creating, filtering, and cleaning the dataset in the supplementary, I believe the main paper should devote more space to this aspect. Specifically, it would benefit from a deeper quantitative and qualitative analysis of data quality and its impact on model performance.

   Additionally, since the cleaning process involves several critical steps, it would be valuable to include analyses or ablation studies illustrating the effect and necessity of each step. This would help readers understand whether the current pipeline can reliably produce high-quality part annotations, and what the remaining challenges are in the data curation process.

   I acknowledge that the proposed network architecture is not unique and could be replaced by alternative designs. Therefore, more insights into data curation would significantly strengthen the paper and benefit future research, as the core challenge in this field lies more in *data quality and scalability* than in model design. Currently, the paper emphasizes the model side while underrepresenting the data aspect.

3. **Further experiments on data scale and quality.**
   I would like to see additional experiments exploring the influence of data quantity and quality, such as scaling curves with varying training data sizes. The current baselines are trained with significantly less data than the proposed method, which makes it unclear whether the improvements stem from the model architecture or simply from the larger training dataset. It would be important to evaluate whether the proposed model still outperforms when trained on the same amount of data as the baselines.

**Questions:**

1. **Data release.**
   Will you release the data processing scripts and the processed datasets with part annotations to facilitate future research and reproducibility?

2. **Post-processing hyperparameters.**
   What hyperparameters are involved in the post-processing stage (e.g., NMS)? How were these parameters determined? How do you control the granularity of the “automatic segmentation,” and how do these parameters affect the final results? Including an ablation study on these factors would strengthen the paper.

3. **Title clarification.**
   Would you consider revising the current title? There are already several works doing similar things,  *“Native 3D Part Segmentation,”* and *P³-SAM* does not appear to be the first in this line.

4. **Connectivity information.**
   Could you elaborate on how connectivity information is incorporated into *P³-SAM*?

5. **Hierarchical part segmentation.**
   Even after reading the supplementary material, the description of the *Hierarchical Part Segmentation* module remains unclear. Could you provide more details on how this module operates? For example, how do you obtain part segmentations with very fine granularity?

6. **Watertightness influence.**
   I do not fully understand why the distinction between watertight and non-watertight meshes affects the network, given that the model takes point clouds as input. Could you clarify this effect?

7. **Normal quality impact.**
   If the input normals are of poor quality (e.g., flipped faces or inconsistent directions), how does that affect the final segmentation performance?

8. **Clarification on Figure 6.**
   For Figure 6, could you explain the version without flood fill? Why it looks different than other subfigures?

9. **Runtime and efficiency.**
   Regarding the reported timing, does it correspond to the inference time for a single-point prompt, or to the runtime of the automatic segmentation process that samples multiple point prompts? How many point prompts are typically used for automatic segmentation, and how does this number affect the overall runtime and efficiency?

---

> ### Author Response · Authors · 2025-11-20
>
> We deeply value the reviewers' recognition of our efforts and their detailed feedback, as these insights are pivotal in strengthening the validity and significance of our findings. Here are our responses:
>
> > **Weakness1:** Method novelty and comparison with existing work.
>
> We did draw inspiration primarily from the concept of prompt-based segmentation in SAM, but instead of mechanically replicating it, we thoroughly integrated this idea with the characteristics of point clouds and the functionalities required for automatic full segmentation tasks to design our network. For instance:
> - **Feature Extractor**: We adopted a SOTA network architecture and pretrained weights that are specially designed for point clouds,
> - **Promp**t: We discarded the complex prompt design and prompt training, and instead adopted a single prompt point to focus on 3D object full segmentation.
> - **Mask Decoder**: We developed a unique two-stage segmentation head that only uses an MLP, without employing the Two Way Transformer as in SAM.
> - **Data**: There is no data engine in our method. We directly train our model on our large-scale preprocessed native 3D data.
>
> These are all distinctive and novel designs of ours.
>
> We have revised our description regarding the differences between our method and PointSAM in Section A.2.3 of the appendix, where we provide an explanation of the innovative aspects, advantages, and distinctions of our approach compared to PointSAM from various perspectives:
>
> - **Motivation**: While PointSAM focuses primarily on interactive segmentation tasks, our method is specifically designed for automatic full segmentation. PointSAM does not provide a comprehensive solution for full segmentation, whereas our approach aims to deliver an efficient and effective method for automatically segmenting entire datasets without requiring user interaction.
>
> - **Network Architecture**: PointSAM adapts the network structure of SAM to build a model suitable for point clouds. It utilizes FPS to extract key points and groups surrounding points as tokens, directly employing SAM's architecture as the segmentation network. However, extending 2D network structures to 3D is not always effective. Therefore, unlike PointSAM, we employ state-of-the-art methods for point cloud feature extraction as our feature extraction module. To simplify model training and inference, we removed the complex prompt encoding and adopted a single prompt point. To enhance mask quality, we also introduced a two-stage segmentation module.
>
> - **Data**: Our method does not have a data engine; we directly train on our large-scale preprocessed native 3D data. In contrast, PointSAM constructs a data engine that leverages prior knowledge from 2D SAM, which inevitably introduces inconsistencies from 2D.
>
> - **Our Advantages**:
>     - Our method is a fully native 3D segmentation approach that supports both interactive segmentation and automatic full segmentation.
>     - Our method is trained on a large-scale preprocessed native 3D dataset of 3.7 million samples, avoiding the introduction of 2D segmentation priors.
>     - Our method exhibits strong generalization, robustness, and efficiency, outperforming existing methods and achieving state-of-the-art results across multiple datasets and tasks.

---

> > ### Author Response · Authors · 2025-11-20
> >
> > > **Weakness2:** Dataset curation.
> >
> > The construction of our dataset adheres to a few simple principles and filtering criteria mentioned in the paper, and it does not involve any manual intervention, significantly reducing human labor. We acknowledge that this approach introduces some low-quality noisy data, where the segmentation masks are inaccurate, which could impact the model. However, we believe that the sheer volume of data can mitigate this issue, allowing the network to learn an "average" correct segmentation from the large dataset, which is one of the primary benefits of having extensive data. Below is a detailed description and analysis of our data filtering process:
> > 1. Checking for Connected Components. Designers and artists often create models by constructing individual components separately and then assembling them into a complete object. This component-based construction method can be leveraged to obtain component-wise segmentation data, which forms an indispensable foundation for building our dataset.
> > 2. Compute Area and Occupied Voxels. First, we calculate the area of each connected component's triangular facets and their overall proportion. Then, we divide the space into voxels with a resolution of 128 and compute the occupied voxels for each component. When two components have overlapping occupied voxels, they are considered adjacent, and an adjacency matrix is constructed. This step forms the basis for subsequently merging extremely small connected components.
> > 3. Merge Extremely Small Connected Components. For each connected component, if its area proportion is less than 1%, it is merged with the largest adjacent part. If there are no adjacent parts, no action is taken. This step is repeated until all possible merges are completed. This process helps avoid mask distribution imbalances caused by extremely small parts, thereby improving the utilization of 3D models. A large number of small parts can generate many masks without significant actual area, which can negatively impact training. Without merging, much of the data would be unusable. Additionally, very small parts are not conducive to subsequent watertight processing. Since the merged parts are relatively small, they have minimal impact on the overall quality of most masks. Although some low-quality erroneous masks may still be generated, we believe that the large volume of data can mitigate the effects of these noisy data.
> > 4. Filter Out Data with Too Few or Too Many Components. After merging, we filter out data where the number of parts is less than 2 or greater than 50. Data containing only a single part do not involve object segmentation and need to be filtered out. Data with an excessive number of parts often lack reasonable semantic meaning for each part and can lead to over-segmentation, resulting in a large number of unnecessary parts. For example, a car data containing thousands of parts typically consist of surface patches that do not have meaningful semantic significance.
> > 5. Filter Out Imbalanced Data. When an object contains a very large part, i.e., one part occupies more than 85% of the total area, we filter out this object. Additionally, if an object contains a large number of independent small parts, where the total area of parts with less than 1% area proportion (which could not be merged in step 3) exceeds 10%, we also filter out this object. This step is aimed at removing imbalanced masks, specifically those with extremely large actual regions and those with extremely small actual regions.
> >
> > The primary purpose of our data processing and filtering is to improve data utilization and generate balanced mask distributions. We have revised the above descriptions and analyses in Section A.3.1 of the appendix.

---

> > > ### Author Response · Authors · 2025-11-20
> > >
> > > > **Weakness3:** Further experiments on data.
> > >
> > > Although the construction of the dataset does not require human intervention, performing an ablation study on the data cleaning process and data quality is still resource-intensive and challenging to execute. We conducted experiments on an ablated dataset without watertight data, which also serves as a form of ablation study focusing on data quality. We trained our method on a dataset that did not include watertight data and evaluated it on 3D generated models. The results are presented in Figure 12 the appendix.
> > > Since these generated models are watertight, while this method was not trained on watertight data, the network fails to segment these models effectively, resulting in only one or two parts being identified. Therefore, the importance of watertight data in our method's performance is significant.
> > >
> > >
> > > Following the reviewers' suggestions, we conducted experiments to evaluate the impact of dataset scale on network performance by varying the amount of training data used.
> > > We trained our model on datasets of 100K, 500K, 1M, and the full dataset(~2M). To ensure a fair comparison, all models were trained on 32 H20 GPUs with identical settings, including the same batch size, for 40K steps. We evaluated the "Full Segmentation without Connectivity" task on the PartObj-Tiny dataset. The results are presented in the table below.
> > >
> > > | Data Num | 100K | 500K | 1M | Full |
> > > | ---- |  ---- |  ---- |  ---- |  ---- |
> > > | IoU | 56.42 | 56.49 | 56.56 | 57.06 |
> > >
> > > It is evident that reducing the dataset size negatively impacts model performance. In addition to quantitative evaluations, we also conducted a qualitative analysis of the models' performance on 3D generated data, as shown in Figure 14 of the appendix. Reducing the dataset size significantly affects the model's generalization ability, leading to poorer segmentation results on 3D generated data. This experiment supports our hypothesis that large-scale datasets can mitigate some of the issues caused by imperfect data construction.
> > >
> > > We would like to express our gratitude once again for your recognition of our contributions in data. However, our methodological contributions are also significant and should not be overlooked. Following your suggestions, we trained different methods on our full dataset and evaluated their performance. Specifically, we trained PartField, PointSAM, and an ablated method to validate the effectiveness and innovativeness of our approach. The ablated method "Sonata+CL" combines feature extraction from Sonata with the contrastive learning loss from PartField. The ablated method "Sonata+CL" differs from PartField primarily in its feature extraction module, which has been replaced with the feature extraction mechanism from Sonata. Compared to our full method, "Sonata+CL" lacks the specialized segmentation head that we introduced.  The comparison results on vsrious tasks are presented in the tables below.
> > >
> > > | Full Seg. w/ C.| Org. Data| Ours Data |
> > > | ---- |  ---- |  ---- |
> > > | PartField | 79.18 | 78.33 |
> > > | Sonata+CL | - | 76.41 |
> > > | Ours | - | 81.14 |
> > >
> > > | Full Seg. w/o C.| Org. Data| Ours Data |
> > > | ---- |  ---- |  ---- |
> > > | PartField | 53.93 | 55.62 |
> > > | Sonata+CL | - | 50.02 |
> > > | Ours | - | 59.88 |
> > >
> > > | Inter. Seg. | Org. Data| Ours Data |
> > > | ---- |  ---- |  ---- |
> > > | PointSAM | 27.91 | 28.64 |
> > > | Ours | - | 51.23 |
> > >
> > > Even when the comparison methods were trained on our full dataset, their performance did not surpass those of our method, demonstrating the effectiveness and innovativeness of our approach. Notably, PartField's performance on the "Full Segmentation with Connectivity" task showed a slight decline, further highlighting the stability of our method when trained on large-scale data.
> > > The performance of "Sonata+CL" did not surpass that of PartField or our method, further demonstrating the uniqueness and innovativeness of our approach, particularly our two-stage segmentation module.
> > > Thus, the superiority of our method stems not only from the quality and scale of the data but also from our unique and innovative network design.

---

> > > > ### Author Response · Authors · 2025-11-20
> > > >
> > > > > **Q1:** Data release.
> > > >
> > > > Yes, we will make our dataset and data construction scripts publicly available.
> > > >
> > > > > **Q2:** Post-processing hyperparameters.
> > > >
> > > > Our automatic post-processing involves two hyperparameters: the number of prompt points and the overlap threshold used in the NMS process to determine whether two masks overlap. We determined the optimal settings by comparing the segmentation performance of our method on 3D models under different configurations. During the automatic segmentation process, the network "self-determines" the granularity of objects without human intervention, similar to how NMS determines the number of objects in image object detection.
> > > > Following your sugessts, we conducted ablation studies on these two hyperparameters evaluated on the "Full Segmentation without Connectivity" task of the PartObj-Tiny dataset. The results are presented in the tables below.
> > > >
> > > > | NMS Threshold | 0.70|	0.80	|0.85|	0.90|	0.95|
> > > > | ---- |---- |---- |---- |---- |---- |
> > > > | IoU|	59.9|	59.94|	59.87|	59.88|	59.55|
> > > >
> > > > |Prompt Number|	10|	50|	100|	200|	400|	600|	800|
> > > > | ---- |---- |---- |---- |---- |---- |---- |---- |
> > > > |IoU|	44.83|	56.44|	57.73|	58.96|	59.88|	59.90|	60.31|
> > > > |Time(Second)|	0.581|	1.569|	2.416|	4.203|	8.214|	13.308|	17.016|
> > > >
> > > > The default values for these two hyperparameters are 0.9 for the NMS overlap threshold and 400 for the number of prompt points. It is evident that the method's performance is not sensitive to the NMS threshold. This indicates the accuracy of our approach, meaning that the segmentation masks for the same part under different numbers of prompt points are quite similar, and the masks of different parts rarely overlap. Even with smaller thresholds, there is no significant risk of incorrectly determining whether two masks overlap.
> > > >
> > > > As the number of prompt points increases, the performance of our method gradually improves. This is because a higher number of prompt points can better cover all parts, allowing the method to segment as many parts as possible. However, this also leads to an increase in computational time. Therefore, we opted for a balanced setting of 400 prompt points, which provides a good trade-off between performance and efficiency.
> > > > We also demonstrate the impact of different numbers of prompt points on segmentation quality in Figure 15 of the appendix. This figure visually illustrates that as the number of prompt points increases, the number of parts identified and segmented also increases, with more smaller parts being successfully segmented out.
> > > >
> > > > > **Q3:** Title clarification
> > > >
> > > > To the best of our knowledge, currently published papers do not include any part segmentation methods that operate entirely without 2D data or 2D priors. We are indeed the first to achieve part segmentation using only 3D data. Given that traditional point cloud segmentation methods also use only 3D data, we will consider modifying our title to further emphasize our method's zero-shot capability for segmenting arbitrary objects and parts, thereby distinguishing it from traditional approaches.
> > > >
> > > > > **Q4:** Connectivity information.
> > > >
> > > > Connectivity information includes the connections between mesh vertices and the connected components within the mesh. Specifically, we assign a single label to each connected component after voting, which significantly improves segmentation quality and mask accuracy on the PartObj-Tiny dataset. Additionally, the connectivity between mesh vertices helps in identifying neighboring points, thereby enabling the removal of smaller noise masks. However, in practical applications, connected components may not exist in certain 3D-generated objects. Therefore, evaluating the method's ability to perform accurate segmentation without connectivity information is crucial to fully demonstrate its segmentation performance.
> > > >
> > > > > **Q5:** Hierarchical part segmentation.
> > > >
> > > > We begin by segmenting the mesh to identify each part using our P3-SAM. For each identified part, we compute its average feature by finding the corresponding sampled points and extracting their features from the feature extraction module. We then take the mean of these features as the representative feature for that part. Subsequently, we apply hierarchical clustering on these average features to obtain a hierarchical segmentation result.
> > > > Our method automatically segments objects, determining the number of parts autonomously. Therefore, it does not inherently support further subdividing existing segmentation results. However, with simple modifications, our method can achieve fine-grained partitioning. For instance:
> > > > - During mask prediction, instead of selecting the mask with the highest IoU, we could output the smallest mask among the predictions.
> > > > - Or we could recursively apply our method to subdivide parts. Specifically, after initially identifying a part, we can reapply our segmentation algorithm to that part to further divide it into smaller components.

---

> > > > > ### Author Response · Authors · 2025-11-20
> > > > >
> > > > > > **Q6:** Watertightness influence.
> > > > >
> > > > > Even though point clouds are relatively sparse representations, sampling from non-watertight meshes retains the characteristics of those meshes. We visually demonstrate the differences between watertight and non-watertight data in Figure 12 of the appendix. For example, in a human body model, the leg and body parts may intersect, leading to overlapping sampled point clouds. This results in discontinuous distributions of points and normals in these regions. In contrast, watertight meshes have connected leg and body parts, resulting in continuous distributions of points and normals. The discontinuities in point cloud and normal distributions provide more distinct features, making it easier for networks to segment them without truly learning the semantic features of the object. Our experiments also show that segmentation metrics are lower for watertight data, confirming that segmenting watertight data is more challenging.
> > > > > As mentioned in W3, we removed the watertight data from our dataset and trained the model, which resulted in significantly poorer segmentation performance on the 3D generation model. This highlights the impact of watertight data on network performance and underscores its critical importance.
> > > > >
> > > > > > **Q7** Normal quality impact
> > > > >
> > > > > Our method incorporates data augmentation techniques, including randomly removing normals and adding noise to point clouds and normals. This enhances our method's robustness in handling scenarios where normals are missing or noisy. To validate this, we conducted experiments on the "Full Segmentation without Connectivity" task using the PartObj-Tiny dataset. Following the augmentation methods detailed in Section A.3.2 of the appendix, we added random noise to the point clouds and normals and also removed the normals entirely. The results are presented in the table below.
> > > > >
> > > > > |Noise $s_{max}$|	0|	0.001|	0.005|	0.01|	0.05|	0.1|
> > > > > | ---- | ---- | ---- | ---- | ---- | ---- | ---- |
> > > > > |w/ Normal|	59.88|	59.34|	58.41|	57.34|	31.75|	21.13|
> > > > > |w/o Normal|	57.56|	57.22|	56.04|	55.02|	29.25|	20.70|
> > > > >
> > > > > The results indicate that our method maintains SOTA performance when the random noise is less than 0.01. Additionally, the performance does not significantly degrade in the absence of normals. These results demonstrate the excellent robustness of our approach.
> > > > >
> > > > > > **Q8:** Clarification on Figure 6.
> > > > >
> > > > > Since some models (e.g., 3D generative models) have a high number of faces (approximately 500k compared to 100k sampled points), it is not possible to sample point clouds on every face. As a result, many faces are not sampled, leading to unlabeled faces during the labeling process. To address this issue, a flooding strategy is employed to assign labels to these faces.
> > > > >
> > > > > > **Q9:** Runtime and efficiency.
> > > > >
> > > > > The reported time of 8 seconds is for segmenting with 400 prompt points. The automatic segmentation uses 400 prompt points. The number of prompt points and the processing time have a linear relationship; however, typically 400 points are sufficient, as additional points beyond this only increase the number of redundant masks. Detailed runtime and prompt point quantity relationships can be seen in the ablation study in Q2.

---

> > > > > > ### Author Response · Authors · 2025-11-27
> > > > > > **Rebuttal Feedback Request**
> > > > > >
> > > > > > Thank you for your insightful comments and dedication to evaluating our manuscript. We have submitted a detailed rebuttal addressing all your points. We kindly request your feedback on our responses at your earliest convenience, and we would be pleased to discuss any points further with you. Your guidance remains invaluable as we refine our work, and we sincerely appreciate your time and expertise.

---

### Author Response · Authors · 2025-11-20

We sincerely thank the reviewers for their esteemed recognition, dedicated efforts, and invaluable feedback. Your comments are instrumental in refining our manuscript, and we deeply value your insights. Detailed responses to each of your points have been provided directly beneath every reviewer's remarks.
We have revised the manuscript, with all modifications clearly marked in blue font. The qualitative experimental visualizations are available in the corresponding sections of the paper.
We eagerly await your feedback on our replies and appreciate your continued guidance in strengthening our work.

---

### Meta-Review · Area_Chair_cvNE · 2025-12-08

**Summary:**

Reviewers agreed the paper presents strong results and a valuable large-scale dataset, but they raised consistent concerns about limited methodological novelty. The approach was viewed as closely following the SAM framework with only minor architectural differences (e.g. exchanging the feature extractor with Sonata), and several reviewers questioned whether the performance gains stem primarily from the dataset size and pretrained models rather than meaningful new ideas. In addition, key components—such as the multi-head predictor—were not analyzed in depth, leaving unclear what insight they provide. Some reviewers also noted limited evaluation breadth and insufficient validation of the dataset curation process.

**Reviewer Concerns:**

The rebuttal addressed several points raised during the review. The authors expanded the experimental analysis, including additional evaluations on 3DCOMPAT++, ablations on dataset size, prompt sampling, and normal/noise robustness. They also clarified aspects of the data construction pipeline and provided more detailed responses regarding implementation choices.

Despite the additional experiments and clarifications, the core concern regarding methodological novelty remains unresolved. In my opinion, the rebuttal did not convincingly demonstrate that the architectural contribution — particularly the multi-head predictor — represents a meaningful conceptual or technical advance beyond existing approaches. Related to this, the question of whether performance gains are primarily due to dataset scale and pretrained models rather than model design also remain not sufficiently addressed. Finally, although the rebuttal clarified the data pipeline, concerns about dataset quality, heuristics for merging small parts, and potential biases introduced during curation were not fully resolved, as no quantitative validation or error analysis of the annotations was provided.

**Reviewer Scores:**

Reviewer imJU (score 4) raised the strongest concerns about novelty, dataset validation, and insufficient analysis of key design choices. The core issue emphasized, i.e. the method novelty, remains unresolved in my opinion. The results still indicate, that the model largely depends on the dataset and the pre-trained Sonata model, and only makes relatively small modifications on top. If the multi-head predictor is indeed the core contribution, it should be studied in more detail and analyzed why exactly it leads to better performance, i.e. what is the core insight? At this stage, this insight remains unclear to me, therefore I think the reviewer is most likely to retain the original score 4.

Reviewer CnDT (score 6) was initially on the positive side, but also highlighted a key limitation - the highly heuristic part annotation in the dataset, which was not convincingly addressed. Therefore, I think the reviewer is most likely to either retain or slightly lower the score.

Reviewer cf8P (score 8) was the most positive and primarily requested broader evaluation and cross-shape consistency analysis. The rebuttal directly addressed these requests with new experiments. The reviewer would probably retain the score 8.

Reviewer UkFr (score 4) was mainly concerned about the methodological innovation (similar to reviewer imJU). As I stated already, this was not addressed in sufficient depth, and therefore the reviewer would probably retain the score 4.

---

### Decision · Program_Chairs · 2026-01-26

Reject